# A Hierarchical Bayesian Model for Few-Shot Meta Learning

**Minyoung Kim**[1] & **Timothy M. Hospedales**[1,2]
[1]Samsung AI Center Cambridge, UK          [2]University of Edinburgh, UK
mikim21@gmail.com                          t.hospedales@ed.ac.uk

## Abstract

We propose novel model parametrisation and inference algorithm in a hierarchical Bayesian model for the few-shot meta learning problem. We consider episode-wise random variables to model episode-specific generative processes, where these local random variables are governed by a higher-level global random variable. The global variable captures information shared across episodes, while controlling how much the model needs to be adapted to new episodes in a principled Bayesian manner. Within our framework, prediction on a novel episode/task can be seen as a Bayesian inference problem. For tractable training, we need to be able to relate each local episode-specific solution to the global higher-level parameters. We propose a Normal-Inverse-Wishart model, for which establishing this local-global relationship becomes feasible due to the approximate closed-form solutions for the local posterior distributions. The resulting algorithm is more attractive than the MAML in that it does not maintain a costly computational graph for the sequence of gradient descent steps in an episode. Our approach is also different from existing Bayesian meta learning methods in that rather than modeling a single random variable for all episodes, it leverages a hierarchical structure that exploits the local-global relationships desirable for principled Bayesian learning with many related tasks.

## 1 Introduction

Few-shot learning (FSL) aims to emulate the human ability to learn from few examples (Lake et al., 2015). It has received substantial and growing interest (Wang et al., 2020b) due to the need to alleviate the notoriously data intensive nature of mainstream supervised deep learning. Approaches to FSL are all based on some kind of knowledge transfer from a set of plentiful source recognition problems to the sparse data target problem of interest. Existing approaches are differentiated in terms of the assumptions they make about what is task agnostic knowledge that can be transferred from the source tasks, and what is task-specific knowledge that should be learned from the sparse target examples. For example, the seminal MAML (Finn et al., 2017) and ProtoNets (Snell et al., 2017) respectively assume that the initialisation for fine-tuning, or the feature extractor for metric-based recognition should be transferred from source categories.

One of the most principled and systematic ways to model such sets of related problems are hierarchical Bayesian models (HBMs) (Gelman et al., 2003). The HBM paradigm is widely used in statistics, but has seen relatively less use in deep learning, due to the technical difficulty of bringing hierarchical Bayesian modelling to bear on deep learning. HBMs provide a powerful way to model a set of related problems, by assuming that each problem has its own parameters (e.g, the neural networks that recognise cat vs dog, or car vs bike), but that those problems share a common prior (the prior over such neural networks). Data-efficient learning of the target tasks is then achieved by inferring the prior based on source tasks, and using it to enhance posterior learning over the target task parameters.

A Bayesian learning treatment of FSL would be appealing due to the overfitting resistance provided by Bayesian Occam's razor (MacKay, 2003), as well as the ability to improve calibration of inference so that the model's confidence is reflective of its probability of correctness — a crucial property in mission critical applications (Guo et al., 2017). However the limited attempts that have been made to exploit these tools in deep learning have either been incomplete treatments that only model a single Bayesian layer within the neural network (Zhang et al., 2021; Gordon et al., 2019), or else fail to scale up to modern neural architectures (Finn et al., 2018; Yoon et al., 2018).

In this paper we present the first complete hierarchical Bayesian learning algorithm for few-shot deep learning. Our algorithm efficiently learns a prior over neural networks during the meta-train phase, and efficiently learns a posterior neural network during each meta-test episode. Importantly, our learning is architecture independent. It can scale up to the state-of-the-art backbones including ViTs (Dosovitskiy et al., 2021), and works smoothly with any few-shot learning architecture – spanning simple linear decoders (Finn et al., 2017; Snell et al., 2017), to those based on sophisticated set-based decoders such as FEAT (Ye et al., 2020) and CNP(Garnelo et al., 2018)/ANP(Kim et al., 2019). We show empirically that our HBM provides improved performance and calibration in all of these cases, as well as providing clear theoretical justification.

Our analysis also reveals novel links between seminal FSL methods such as ProtoNet (Snell et al., 2017), MAML (Finn et al., 2017), and Reptile (Nichol et al., 2018), all of which are different special cases of our framework despite their very different appearance. Interestingly, despite its close relatedness to MAML-family algorithms, our Bayesian learner admits an efficient closed-form solution to the task-specific and task-agnostic updates that does not require maintaining the computational graph for reverse-mode backpropagation. This provides a novel solution to a famous meta-learning scalability bottleneck. **In summary, our contributions include**: (i) The first complete hierarchical Bayesian treatment of the few-shot deep learning problem, and associated theoretical justification. (ii) An efficient algorithmic learning solution that can scale up to modern architectures, and plug into most existing neural FSL meta-learners. (iii) Empirical results demonstrating improved accuracy and calibration performance on both classification and regression benchmarks.

## 2 PROBLEM SETUP

Let $p(\mathcal{T})$ be the (unknown) task/episode distribution, where each task $\mathcal{T} \sim p(\mathcal{T})$ is defined as a distribution $p_\mathcal{T}(x, y)$ for data $(x, y)$ where $x$ is input and $y$ is target. For training, we have a large number of episodes, $\mathcal{T}_1, \mathcal{T}_2, \ldots, \mathcal{T}_N \sim P(\mathcal{T})$ sampled i.i.d., but we only observe a small number of labeled samples from each episode, denoted by $D_i = \{(x_j^i, y_j^i)\}_{j=1}^{n_i} \sim p_{\mathcal{T}_i}(x, y)$, where $n_i = |D_i|$ is the number of samples in $D_i$. The goal of the learner, after observing the training data $D_1, \ldots, D_N$ from a large number of different tasks, is to build a predictor $p^*(y|x)$ for novel unseen tasks $\mathcal{T}^* \sim p(\mathcal{T})$. We will often abuse the notation, e.g., $i \sim \mathcal{T}$ refers to the episode $i$ sampled, i.e., $D_i \sim p_{\mathcal{T}_i}(x, y)$ where $\mathcal{T}_i \sim p(\mathcal{T})$. At the test time we are allowed to have some hints about the new test task $\mathcal{T}^*$, in the form of a few labeled examples from $\mathcal{T}^*$, also known as the *support set* denoted by $D^* \sim P_{\mathcal{T}^*}(x, y)$.

In Bayesian perspective, the goal is to infer the posterior distribution with all training episodes and a test support set as *evidence*, i.e., $p(y|x, D^*, D_{1:N})$. A major computational challenge, compared to conventional Bayesian learning, is that the training episodes (evidence) may not be stored/replayed/revisited.

Figure 1: (a) IID episodes. (b) Individual episode. (c): FSL as Bayesian inference (grey nodes = *evidences*, red = *target* to infer). $D^*$ = support set for test episode.

## 3 MAIN APPROACH

We introduce two types of latent random variables, $\phi$ and $\{\theta_i\}_{i=1}^N$. Each episode $i$ uses neural network weights $\theta_i$ for modeling the data $D_i$ ($i = 1, \ldots, N$). Specifically, $D_i$ is generated (input $x$ given and only $p(y|x)$ modeled) by $\theta_i$ as in the likelihood model in (1). The variable $\phi$ can be viewed as a globally shared variable that is responsible for linking the individual episode-wise parameters $\theta_i$. We assume conditionally independent and identical priors, $p(\{\theta_i\}_i|\phi) = \prod_i p(\theta_i|\phi)$. Thus the prior for the latent variables $(\phi, \{\theta_i\}_{i=1}^N)$ is formed in a hierarchical manner as follows. (For background on Bayesian modeling, refer to (Murphy, 2022).)

(Prior) $p(\phi, \theta_{1:N}) = p(\phi)\prod_{i=1}^N p(\theta_i|\phi)$,      (Likelihood) $p(D_i|\theta_i) = \prod_{(x,y)\in D_i} p(y|x, \theta_i)$      (1)

where $p(y|x, \theta_i)$ is a conventional neural network model. See the graphical model in Fig. 1(a).

Given the training data $\{D_i\}_{i=1}^N$, the posterior is $p(\phi, \theta_{1:N}|D_{1:N}) \propto p(\phi) \prod_{i=1}^N p(\theta_i|\phi)p(D_i|\theta_i)$, and we approximate it with variational inference. That is, $q(\phi, \theta_{1:N}; L) \approx p(\phi, \theta_{1:N}|D_{1:N})$ where

$$q(\phi, \theta_{1:N}; L) := q(\phi; L_0) \cdot \prod_{i=1}^N q_i(\theta_i; L_i),$$      (2)

where the variational parameters $L$ consists of $L_0$ (parameters for $q(\phi)$) and $\{L_i\}_{i=1}^N$'s (parameters of $q_i(\theta_i)$'s for episode $i$). Note that although $\theta_i$'s are independent across episodes under (2), they

are differently modeled (note the subscript $i$ in notation $q_i$), reflecting different posterior beliefs originating from heterogeneity of episodic datasets $D_i$.

**Normal-Inverse-Wishart (NIW) model.** We consider NIW distributions for the prior and variational posterior. First, the prior is modeled as a conjugate form of Gaussian–NIW. With $\phi = (\mu, \Sigma)$,

$$p(\phi) = \mathcal{N}(\mu; \mu_0, \lambda_0^{-1}\Sigma) \cdot \mathcal{IW}(\Sigma; \Sigma_0, \nu_0), \quad p(\theta_i|\phi) = \mathcal{N}(\theta_i; \mu, \Sigma), \ \ i = 1, \dots, N, \quad (3)$$

where $\Lambda = \{\mu_0, \Sigma_0, \lambda_0, \nu_0\}$ is the parameters of the NIW. We do not need to pay attention to the choice of values for $\Lambda$ since $p(\phi)$ has vanishing effect on posterior for a large amount of evidence as we will see shortly. Next, our choice of the variational density family for $q(\phi)$ is the NIW, mainly because it admits closed-form expressions in the ELBO function due to the conjugacy, allowing efficient local episodic optimisation, as will be shown. For $q_i(\theta_i)$'s we adopt Gaussians. That is,

$$q(\phi; L_0) := \mathcal{N}(\mu; m_0, l_0^{-1}\Sigma) \cdot \mathcal{IW}(\Sigma; V_0, n_0), \quad q_i(\theta_i; L_i) = \mathcal{N}(\theta_i; m_i, V_i). \quad (4)$$

So, $L_0 = \{m_0, V_0, l_0, n_0\}$ with $V_0$ restricted to be diagonal, and $L_i = \{m_i, V_i\}$. Learning (variational inference) amounts to finding $L_0$ and $\{L_i\}_{i=1}^N$ that makes the approximation $q(\phi, \theta_{1:N}; L) \approx p(\phi, \theta_{1:N}|D_{1:N})$, as tight as possible.

**Variational inference.** The negative marginal log-likelihood (NMLL) has the following upper bound (Appendix B.1 for derivations):

$$-\log p(D_{1:N}) \leq \mathrm{KL}(q(\phi)||p(\phi)) + \sum_{i=1}^N \left( \mathbb{E}_{q_i(\theta_i)}[l_i(\theta_i)] + \mathbb{E}_{q(\phi)}\big[\mathrm{KL}(q_i(\theta_i)||p(\theta_i|\phi))\big] \right) \quad (5)$$

where $l_i(\theta_i) = -\log p(D_i|\theta_i)$ is the negative training log-likelihood of $\theta_i$ in episode $i$. By dividing both sides by $N$, the LHS naturally becomes the *effective episode-averaged NMLL* $-\frac{1}{N}\log p(D_{1:N})$. The first KL term in the RHS, $\frac{1}{N}\mathrm{KL}(q(\phi)||p(\phi))$ diminishes for large $N$. Using $\frac{1}{N}\sum_{i=1}^N f_i \approx \mathbb{E}_{i\sim\mathcal{T}}[f_i]$ for any expression $f_i$, the ELBO learning (approximately) reduces to the following:

$$\min_{L_0, \{L_i\}_{i=1}^N} \mathbb{E}_{i\sim\mathcal{T}}\Big[ \mathbb{E}_{q_i(\theta_i; L_i)}[l_i(\theta_i)] + \mathbb{E}_{q(\phi; L_0)}\big[\mathrm{KL}(q_i(\theta_i; L_i)||p(\theta_i|\phi))\big] \Big]. \quad (6)$$

**Local episodic optimisation (whose solution as a function of global parameters $L_0$).** Note that (6) is challenging due to a large number of optimisation variables $\{L_i\}_{i=1}^N$ and the nature of episode sampling $i \sim \mathcal{T}$. Applying conventional SGD would simply fail since each $L_i$ will never be updated more than once. Instead, we tackle it by finding the optimal solutions for $L_i$'s for fixed $L_0$, thus effectively representing the optimal solutions as functions of $L_0$, namely $\{L_i^*(L_0)\}_{i=1}^N$. Plugging the optimal $L_i^*(L_0)$'s back to (6) leads to the optimisation problem over $L_0$ alone. The idea is just like solving: $\min_{x,y} f(x, y) = \min_x f(x, y^*(x))$ where $y^*(x) = \arg\min_y f(x, y)$ with $x$ fixed.

Note that when we fix $L_0$ (i.e., fix $q(\phi)$), the objective (6) is completely separable over $i$, and we can optimise individual $i$ independently. More specifically, for each $i \geq 1$,

$$\min_{L_i} \mathbb{E}_{q_i(\theta_i; L_i)}[l_i(\theta_i)] + \mathbb{E}_\phi\big[\mathrm{KL}(q_i(\theta_i; L_i)||p(\theta_i|\phi))\big] \quad (7)$$

As the expected KL term in (7) admits a closed form due to NIW-Gaussian conjugacy (Appendix B.2 for derivations), we can reduce (7) to the following optimisation for $L_i = (m_i, V_i)$:

$$L_i^*(L_0) := \arg\min_{m_i, V_i} \left( \mathbb{E}_{\mathcal{N}(\theta_i; m_i, V_i)}[l_i(\theta_i)] - \frac{\log |V_i|}{2} + \frac{n_0}{2}\Big((m_i - m_0)^2/V_0 + \mathrm{Tr}\big(V_i/V_0\big)\Big) \right) \quad (8)$$

with $L_0 = \{m_0, V_0, l_0, n_0\}$ fixed. Here $(m_i - m_0)^2$ and $\cdot/V_0$ are all elementwise operations.

**Quadratic approximation of episodic loss via SGLD.** To find the closed-form solution $L_i^*(L_0)$ in (8), we make quadratic approximation of $l_i(\theta_i) = -\log p(D_i|\theta_i)$. In general, $-\log p(D_i|\theta)$, as a function of $\theta$, can be written as:

$$-\log p(D_i|\theta) \approx \frac{1}{2}(\theta - \overline{m}_i)^\top \overline{A}_i (\theta - \overline{m}_i) + \mathrm{const.}, \quad (9)$$

for some $(\overline{m}_i, \overline{A}_i)$ that are constant with respect to $\theta$. One may attempt to obtain $(\overline{m}_i, \overline{A}_i)$ via Laplace approximation (e.g., the minimiser of $-\log p(D_i|\theta)$ for $\overline{m}_i$ and the Hessian at the minimiser for $\overline{A}_i$). However, this involves computationally intensive Hessian computation. Instead, using the fact that the log-posterior $\log p(\theta|D_i)$ equals (up to constant) $\log p(D_i|\theta)$ when we use uninformative prior $p(\theta) \propto 1$, we can obtain samples from the posterior $p(\theta|D_i)$ using MCMC sampling, especially the stochastic gradient Langevin dynamics (SGLD) (Welling & Teh, 2011), and estimate sample mean and precision, which become $\overline{m}_i$ and $\overline{A}_i$, respectively[1]. Note that this amounts to performing

---

[1]Similar approaches include the stochastic weight averaging (Izmailov et al., 2018; Maddox et al., 2019).

several SGD iterations (skipping a few initial for burn-in), and unlike MAML (Finn et al., 2017) no computation graph needs to be maintained since $(\overline{m}_i, \overline{A}_i)$ are constant. Once we have $(\overline{m}_i, \overline{A}_i)$, the optimisation (8) admits the closed-form solution (Appendix B.4 for derivations),

$$m_i^*(L_0) = (\overline{A}_i + n_0/V_0)^{-1}(\overline{A}_i \overline{m}_i + n_0 m_0/V_0), \qquad V_i^*(L_0) = (\overline{A}_i + n_0/V_0)^{-1}. \qquad (10)$$

Computation in (10) is cheap since all matrices are diagonal.

**Final optimisation.**    Plugging (10) back to (6), the final optimisation is (Appendix B.5 for details):

$$\min_{L_0}\ \mathbb{E}_{i\sim\mathcal{T}}\left[f_i(L_0) + \frac{1}{2}g_i(L_0) + \frac{d}{2l_0}\right]\ \text{ s.t. }\ f_i(L_0) = \mathbb{E}_{\epsilon\sim\mathcal{N}(0,I)}\left[l_i\left(m_i^*(L_0) + V_i^*(L_0)^{1/2}\epsilon\right)\right],$$

$$g_i(L_0) = \log\frac{|V_0|}{|V_i^*(L_0)|} + n_0\mathrm{Tr}\left(V_i^*(L_0)/V_0\right) + n_0\left(m_i^*(L_0) - m_0\right)^2/V_0 - \psi_d\left(\frac{n_0}{2}\right), \quad (11)$$

where $\psi_d(\cdot)$ is the multivariate digamma function and $d = \dim(\theta)$. As $l_0$ only appears in the term $\frac{d}{2l_0}$, the optimal value is $l_0^* = \infty$. We use SGD to solve (11), repeating the two steps: i) Sample $i \sim \mathcal{T}$; ii) $L_0 \leftarrow L_0 - \eta \nabla_{L_0}\left(f_i(L_0) + \frac{1}{2}g_i(L_0)\right)$. Note that $\nabla_{L_0}\left(f_i(L_0) + \frac{1}{2}g_i(L_0)\right)$ is an *unbiased* stochastic estimate for the gradient of the objective $\mathbb{E}_{i\sim\mathcal{T}}[\cdots]$ in (11).

---

**Algorithm 1** Our few-shot meta learning algorithm.

**Initialise:** $L_0 = \{m_0, V_0, n_0\}$ of $q(\phi; L_0)$ randomly.
**for** episode $i = 1, 2, \dots$ **do**
    Perform SGLD iterations on $D_i$ to estimate $(\overline{m}_i, \overline{A}_i)$.
    Compute the episodic minimiser $L_i^*(L_0)$ from (10).
    Update $L_0$ by the gradient of $f_i(L_0) + \frac{1}{2}g_i(L_0)$ as in (11).
**end for**
**Output:** Learned $L_0$.

---

Furthermore, our learning algorithm above (pseudocode in Alg. 1) is fully compatible with the online/batch episode sampling nature of typical FSL. After training, we obtain the learned $L_0$, and the posterior $q(\phi; L_0)$ will be used at the meta test time, where we show in Sec. 3.2 that this can be seen as Bayesian inference as well.

We emphasise that our framework is completely flexible in the choice of the backbone $p(y|x, \theta)$. It could be the popular instance-based network comprised of a feature extractor and a prediction head where the latter can be either a conventional learnable readout head or the parameter-free one like the nearest centroid classifier (NCC) in ProtoNet (Snell et al., 2017), i.e., $p(D|\theta) = p(Q|S, \theta)$ where $D = S \cup Q$ and $p(y|x, S, \theta)$ is the NCC prediction with support $S$. We can also adopt the set-based networks (Ye et al., 2020; Garnelo et al., 2018; Kim et al., 2019) where $p(y|x, S, \theta)$ itself is modeled by a neural net $y = G(x, S; \theta)$ with input $(x, S)$.

### 3.1 INTERPRETATION

We show that our framework unifies seemingly unrelated seminal FSL algorithms into one perspective.

**MAML (Finn et al., 2017) as a special case.**    Suppose we have spiky variational densities, $V_i \to 0$ (constant). The local episodic optimisation (8) reduces to: $\arg\min_{m_i} l_i(\theta_i) + R(m_i)$ where $R(m_i)$ is the quadratic penalty of $m_i$ deviating from $m_0$. One reasonable solution is to perform a few gradient steps with loss $l_i$, starting from $m_0$ to have small penalty ($R = 0$ initially). That is, $m_i \leftarrow m_0$ and a few steps of $m_i \leftarrow m_i - \alpha \nabla l_i(m_i)$ to return $m_i^*(L_0)$. Plugging this into (11) and disregarding the $g_i$ term, leads to the MAML algorithm. Obviously, the main drawback is $m_i^*(L_0)$ is a function of $m_0 \in L_0$ via a full computation graph of SGD steps, compared to our lightweight closed forms (10).

**ProtoNet (Snell et al., 2017) as a special case.**    With $V_i \to 0$, if we ignore the negative log-likelihood term in (8), then the optimal solution becomes $m_i^*(L_0) = m_0$. If we remove the $g_i$ term, we can solve (11) by simple gradient descent with $\nabla_{m_0}(-\log p(D_i|m_0))$. We then adopt the NCC head and regard $m_0$ as sole feature extractor parameters, which becomes exactly the ProtoNet update.

**Reptile (Nichol et al., 2018) as a special case.**    Instead, if we ignore all penalty terms in (8) and follow our quadratic approximation (9) with $V_i \to 0$, then $m_i^*(L_0) = \overline{m}_i$. It is constant with respect to $L_0 = (m_0, V_0, n_0)$, and makes the optimisation (11) very simple: the optimal $m_0$ is the average of $\overline{m}_i$ for all tasks $i$, i.e., $m_0^* = \mathbb{E}_{i\sim\mathcal{T}}[\overline{m}_i]$ (we ignore $V_0$ here). Note that Reptile ultimately finds the exponential smoothing of $m_i^{(k)}$ over $i \sim \mathcal{T}$ where $m_i^{(k)}$ is the iterate after $k$ SGD steps for task $i$. This can be seen as an online/running estimate of $\mathbb{E}_{i\sim\mathcal{T}}[\overline{m}_i]$.

### 3.2 META TEST PREDICTION AS BAYESIAN INFERENCE

At meta test time, we need to be able to predict the target $y^*$ of a novel test input $x^* \sim \mathcal{T}^*$ sampled from the unknown distribution $\mathcal{T}^* \sim p(\mathcal{T})$. In FSL, we have the test support data $D^* = \{(x, y)\} \sim \mathcal{T}^*$. The

Table 1: Three competing Bayesian models for the toy experiment. (Fig. 3 for graphical models)

| Model I | Model II | Model III (Ours) |
|---|---|---|
| $y = \theta_i^\top x + \beta_i + \epsilon_y,$ | $y = \theta^\top x + \beta + \epsilon_y,$ | $y = \theta_i^\top x + \beta_i + \epsilon_y,$ |
| $p(\theta_i, \beta_i) = \mathcal{N}(\mu, \sigma^2) \ \forall i$ | $p(\theta, \beta) = \mathcal{N}(\mu, \sigma^2)$ | $p(\phi) = \mathcal{N}(m, V), \ p(\theta_i, \beta_i|\phi) = \mathcal{N}(\phi, \sigma^2) \ \forall i$ |

test-time prediction can be seen as a posterior inference problem with *additional evidence* of the support data $D^*$ (Fig. 1(c)). More specifically, $p(y^*|x^*, D^*, D_{1:N}) = \int p(y^*|x^*, \theta) \, p(\theta|D^*, D_{1:N}) \, d\theta$. So, it boils down to $p(\theta|D^*, D_{1:N})$, the posterior given both the test support data $D^*$ and the entire training data $D_{1:N}$. Under our hierarchical model, exploiting conditional independence (Fig. 1(c)), we can link it to our trained $q(\phi)$ as:

$$p(\theta|D^*, D_{1:N}) \approx \int p(\theta|D^*, \phi) \, p(\phi|D_{1:N}) \, d\phi \ \approx \ \int p(\theta|D^*, \phi) \, q(\phi) \, d\phi \ \approx \ p(\theta|D^*, \phi^*), \quad (12)$$

where in the first approximation in (12) we disregard the impact of $D^*$ on the higher-level $\phi$ given the joint evidence, i.e., $p(\phi|D^*, D_{1:N}) \approx p(\phi|D_{1:N})$, due to dominance of $D_{1:N}$ compared to smaller $D^*$. We use the delta function approximation in the last part of (12) with the mode $\phi^*$ of $q(\phi)$, where $\phi^* = (\mu^*, \Sigma^*)$ has a closed form $\mu^* = m_0, \Sigma^* = V_0/(n_0+d+2)$.

Next, since $p(\theta|D^*, \phi^*)$ involves difficult marginalisation $p(D^*|\phi^*) = \int p(D^*|\theta)p(\theta|\phi^*)d\theta$, we adopt variational inference, introducing a tractable variational distribution $v(\theta) \approx p(\theta|D^*, \phi^*)$. With the Gaussian family as in the training time (4), i.e., $v(\theta) = \mathcal{N}(\theta; m, V)$ where $(m, V)$ are the variational parameters optimised by ELBO optimisation,

$$\min_{m, V} \ \mathbb{E}_{v(\theta)}[-\log p(D^*|\theta)] + \text{KL}(v(\theta)||p(\theta|\phi^*)) \ \text{ where } \ \phi^* = (m_0, V_0/(n_0+d+2)). \quad (13)$$

The detailed derivations for (13) can be found in Appendix B.6. Once we have the optimised model $v$, our predictive distribution can be approximated by the Monte-Carlo average. $p(y^*|x^*, D^*, D_{1:N}) \approx (1/M_S) \sum_{s=1}^{M_S} p(y^*|x^*, \theta^{(s)})$, where $\theta^{(s)} \sim v(\theta)$ for $s = 1, \ldots, M_S$ samples. which simply requires feed-forwarding $x^*$ through the sampled networks $\theta^{(s)}$ and averaging. Our meta-test algorithm is also summarised in Alg. 2 (Appendix). Note that we have test-time backbone update as per (13), which can make the final $m$ deviate from the learned mean $m_0$. Alternatively, if we drop the first term in (13), the optimal $v(\theta)$ equals $p(\theta|\phi^*) = \mathcal{N}(\theta; m_0, V_0/(n_0+d+2))$. This can be seen as using the learned model $m_0$ with some small random perturbation as a test-time backbone $\theta$.

## 4 TOY EXPERIMENT: WHY HIERARCHICAL BAYESIAN MODEL?

To demonstrate why our hierarchical Bayesian modelling is effective for few-shot meta learning problems, we devise a simple toy synthetic experiment as a proof of concept. We consider a multi-task (Bayesian) linear regression problem. The data pairs $(x \in \mathbb{R}^2, y \in \mathbb{R})$ for each episode $i$ are generated by the following process: $y = (w_{\text{shared}} + \epsilon_w)^\top x + b_{j(i)} + \epsilon_y$ where $w_{\text{shared}}$ is the episode-agnostic shared weight vector $\forall i$, and we have episode-dependent intercept $b_{j(i)}$ – among the three candidates $\{b_1, b_2, b_3\}$, we select $j(i) \sim \{1, 2, 3\}$ uniformly at random for each episode $i$. *Please refer to Appendix. C for full details and derivations.* In this way we ensure that the resulting episodes are not only related to one another through the shared weight vector, but they are differentiated by potentially different intercepts. We sample $N = 40$ episodes for training and 10 episodes for test. Each training episode has $|D_i| = 3$ samples, and the support set at test time also has $|D_*| = 3$ labeled samples.

**Three competing models.** We consider three Bayesian models with different levels/degrees of flexibility and regularisation as outlined in Table 1. **Model I** has episode-wise parameters $(\theta_i, \beta_i)$, thus highly flexible. However, these parameters are all independent across episodes, hence lacking regularisation. **Model II** is a conventional (non-hierarchical) Bayesian model where a single parameter set $(\theta, \beta)$ is shared across episodes, thus too much regularisation with lack of flexibility. **Model III** is our hierarchical Bayesian model which imposes balanced flexibility and regularisation – episode-wise $(\theta_i, \beta_i)$'s allows high flexibility, but unlike Model I, the higher-level variable $\phi$ regularises the episode-specific parameters, and captures the inter-episodic shared information.

**Results.** After training (learning $\mu$ for Model I and II; learning $(m, V)$ for our Model III), at test time, for each of 10 test episodes, we obtain the posterior means of the weights and intercept parameters $\mathbb{E}[\theta, \beta|D_*, D_{1:N}]$ for the three models. They all admit closed-form solutions as detailed in Appendix C, and we predict the outputs of $\sim 50$ unseen test inputs. The mean absolute errors (MAE) averaged over 10 test episodes are: **Model I** $= 2.87$, **Model II** $= 3.13$, and **Model III (ours)** $= 1.28$, clearly showing the superiority of our model to other competing methods. Fig. 2 visualises

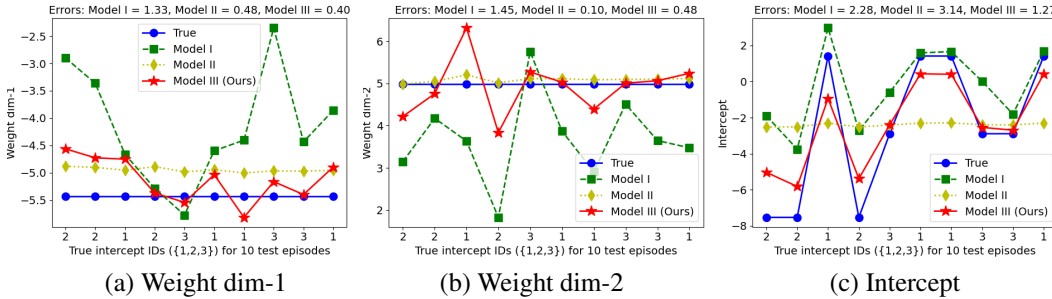

(a) Weight dim-1          (b) Weight dim-2          (c) Intercept

Figure 2: Toy experiments. Visualisation of the learned posterior means compared to the true values (blue-circled). (a) weight dim-1 ($\theta[0]$ vs. $w_{\text{shared}}[0]$), (b) weight dim-2 ($\theta[1]$ vs. $w_{\text{shared}}[1]$) and (c) intercept ($\beta$ vs. $b_{j(*)}$). In each plot, the X-axis shows the indices of the true intercepts sampled, that is, $j(*) \in \{1, 2, 3\}$, for 10 test episodes. In the titles we also report the distances (errors) between the true values and the posterior means for the three methods, averaged over 10 episodes.

the results. First, **Model II**'s posterior means rarely change over test episodes, meaning the impact of test support data $D_*$ is limited. This behavior is expected since the model imposes too much regularisation with little flexibility, and the test prediction is dominated by the mean model obtained from training data $D_{1:N}$. **Model I** exhibits highly sensitive predictions over test episodes, which mainly originates from little regularisation – the posterior is too sensitive to the current episode's support data, thus being vulnerable to overfitting especially when the support data size is small, typical in the few-shot learning. The model failed to capture useful shared information $w_{\text{shared}}$. Our **Model III** balances between these two extremes, imposing proper amount of regularisation and endowing adequate flexibility. Our posterior estimation best extracts the shared episode-agnostic information (the weight parameters fluctuate less over the test episodes), and captures the episode-specific features the most accurately (the estimated intercepts are aligned the best with the true values).

## 5 THEORETICAL ANALYSIS: GENERALISATION ERROR BOUNDS

We offer two theorems for the generalisation error of the proposed model. The first theorem relates the generalisation error to the ultimate ELBO loss (6) that we minimised in our algorithm, and we utilise the recent PAC-Bayes-$\lambda$ bound (Thiemann et al., 2017; Rivasplata et al., 2019). The second theorem is based on the recent regression analysis technique (Pati et al., 2018; Bai et al., 2020). Without loss of generality we assume $|D_i| = n$ for all episodes $i$. We let $(q^*(\phi), \{q_i^*(\theta_i)\}_{i=1}^N)$ be the optimal solution of (6). We leave technical details and the proofs for both theorems in Appendix A.

**Theorem 5.1** (PAC-Bayes-$\lambda$ bound). *Let $R_i(\theta)$ be the generalisation error of model $\theta$ for the task $i$, more specifically, $R_i(\theta) = \mathbb{E}_{(x,y)\sim\mathcal{T}_i}[-\log p(y|x, \theta)]$. As the number of training episodes $N \to \infty$, the following holds with probability at least $1-\delta$ for arbitrary small $\delta > 0$:*

$$\mathbb{E}_{i\sim\mathcal{T}}\mathbb{E}_{q_i^*(\theta_i)}[R_i(\theta_i)] \leq \frac{2\epsilon^*}{n} \quad \text{where } \epsilon^* = \text{the optimal value of (6).} \tag{14}$$

**Theorem 5.2** (Bound derived from regression analysis). *Let $d_H^2(P_{\theta_i}, P^i)$ be the expected squared Hellinger distance between the true distribution $P^i(y|x)$ and model's $P_{\theta_i}(y|x)$ for task $i$. As the number of training episodes $N \to \infty$, the following holds with high probability:*

$$\mathbb{E}_{i\sim\mathcal{T}}\mathbb{E}_{q_i^*(\theta_i)}[d_H^2(P_{\theta_i}, P^i)] \leq O\left(\frac{1}{n} + \epsilon_n^2 + r_n\right) + \lambda^*, \tag{15}$$

*where $\lambda^* = \mathbb{E}_{i\sim\mathcal{T}}[\lambda_i^*]$, $\lambda_i^* = \min_{\theta\in\Theta} ||\mathbb{E}_\theta[y|\cdot] - \mathbb{E}^i[y|\cdot]||_\infty^2$ is the lowest possible regression error within $\Theta$, and $r_n, \epsilon_n$ are decreasing sequences vanishing to 0 as $n$ increases.*

## 6 RELATED WORK

Due to limited space it is infeasible to review all general FSL and meta learning algorithms here. We refer the readers to (Hospedales et al., 2022; Wang et al., 2020a), the excellent comprehensive surveys on the latest techniques. We rather focus on discussing recent Bayesian approaches and their relation to ours. Although several Bayesian FSL approaches have been proposed before, most of them dealt with only a small fraction of the network weights (e.g., a readout head alone) as random variables (Garnelo et al., 2018; Kim et al., 2019; Requeima et al., 2019; Gordon et al., 2019; Patacchiola et al., 2020; Zhang et al., 2021). This considerably limits the benefits from uncertainty modeling of full network parameters.

Table 2: Classification accuracies with standard backbones on *mini*ImageNet and *tiered*ImageNet.

| (a) *mini*ImageNet | | | | | (b) *tiered*ImageNet | | | |
|---|---|---|---|---|---|---|---|---|
| Model | Backbone | 1-Shot | 5-Shot | | Model | Backbone | 1-Shot | 5-Shot |
| AM3 (Xing et al., 2019) | ResNet$^{12}$ | $65.21^{\pm0.49}$ | $75.20^{\pm0.36}$ | | TapNet (Yoon et al., 2019) | ResNet$^{12}$ | $63.08^{\pm0.15}$ | $80.26^{\pm0.12}$ |
| RelationNet2 (Zhang et al., 2020) | ResNet$^{12}$ | $63.92^{\pm0.98}$ | $77.15^{\pm0.59}$ | | RelationNet2 | ResNet$^{12}$ | $68.58^{\pm0.63}$ | $80.65^{\pm0.91}$ |
| MetaOpt (Lee et al., 2019) | ResNet$^{12}$ | $64.09^{\pm0.62}$ | $80.00^{\pm0.45}$ | | MetaOpt | ResNet$^{12}$ | $65.81^{\pm0.74}$ | $81.75^{\pm0.53}$ |
| SimpleShot (Wang et al., 2019) | ResNet$^{18}$ | $62.85^{\pm0.20}$ | $80.02^{\pm0.14}$ | | SimpleShot | ResNet$^{18}$ | $69.09^{\pm0.22}$ | $84.58^{\pm0.16}$ |
| S2M2 (Mangla et al., 2020) | ResNet$^{18}$ | $64.06^{\pm0.18}$ | $80.58^{\pm0.12}$ | | MetaQDA | ResNet$^{18}$ | $69.97^{\pm0.52}$ | $85.51^{\pm0.58}$ |
| MetaQDA (Zhang et al., 2021) | ResNet$^{18}$ | $65.12^{\pm0.66}$ | $80.98^{\pm0.75}$ | | **NIW-Meta (Ours)** | ResNet$^{18}$ | $\mathbf{70.52^{\pm0.19}}$ | $\mathbf{85.83^{\pm0.17}}$ |
| **NIW-Meta (Ours)** | ResNet$^{18}$ | $\mathbf{65.49^{\pm0.56}}$ | $\mathbf{81.71^{\pm0.17}}$ | | LEO (Rusu et al., 2019) | WRN$^{28\text{-}10}$ | $66.33^{\pm0.05}$ | $81.44^{\pm0.09}$ |
| SimpleShot | WRN$^{28\text{-}10}$ | $63.50^{\pm0.20}$ | $80.33^{\pm0.14}$ | | SimpleShot | WRN$^{28\text{-}10}$ | $69.75^{\pm0.20}$ | $85.31^{\pm0.15}$ |
| S2M2 | WRN$^{28\text{-}10}$ | $64.93^{\pm0.18}$ | $83.18^{\pm0.22}$ | | S2M2 | WRN$^{28\text{-}10}$ | $73.71^{\pm0.22}$ | $88.59^{\pm0.14}$ |
| MetaQDA | WRN$^{28\text{-}10}$ | $67.83^{\pm0.64}$ | $84.28^{\pm0.69}$ | | MetaQDA | WRN$^{28\text{-}10}$ | $74.33^{\pm0.65}$ | $89.56^{\pm0.79}$ |
| **NIW-Meta (Ours)** | WRN$^{28\text{-}10}$ | $\mathbf{68.54^{\pm0.26}}$ | $\mathbf{84.81^{\pm0.28}}$ | | **NIW-Meta (Ours)** | WRN$^{28\text{-}10}$ | $\mathbf{74.59^{\pm0.33}}$ | $\mathbf{89.76^{\pm0.23}}$ |

Bayesian approaches to MAML (Finn et al., 2018; Yoon et al., 2018; Ravi & Beatson, 2019; Nguyen et al., 2020) are popular probabilistic extensions of the gradient-based adaptation in MAML (Finn et al., 2017) with known theoretical support (Chen & Chen, 2022). However, they are weak in several aspects to be considered as principled Bayesian methods. For instance, Probabilistic MAML (PMAML) (Finn et al., 2018; Grant et al., 2018) has a similar hierarchical graphical model structure as ours, but their learning algorithm considerably deviates from the original variational inference objective. Unlike the original derivation of the KL term measuring the divergence between the posterior and prior on the task-specific variable $\theta_i$, namely $\mathbb{E}_{q(\phi)}[\text{KL}(q_i(\theta_i|\phi)||p(\theta_i|\phi))]$ as in (5), in PMAML they measure the divergence on the global variable $\phi$, aiming to align the two adapted models, one from the support data only $q(\phi|S_i)$ and the other from both support and query $q(\phi|S_i, Q_i)$. VAMPIRE (Nguyen et al., 2020) incorporates uncertainty modeling to MAML by extending MAML's point estimate to a distributional one that is learned by variational inference. However, it inherits all computational overheads from MAML, hindering scalability. The BMAML (Yoon et al., 2018) is not a hierarchical Bayesian model, but aims to replace MAML's gradient-based *deterministic* adaptation steps by the *stochastic* counterpart using the samples (called particles) from $p(\theta_i|S_i)$, thus adopting stochastic ensemble-based adaptation steps. If a single particle is used, it reduces exactly to MAML. Thus existing Bayesian approaches are not directly related to our hierarchical Bayesian perspective. A related but different line of research studied Bayesian neural processes (Volpp et al., 2023; 2021; Qi Wang, 2020; Garnelo et al., 2018) by treating the support set embedding as random variates.

# 7 EVALUATION

## 7.1 FEW-SHOT CLASSIFICATION

**Standard benchmarks with ResNet backbones.** For standard benchmark comparison using the popular ResNet backbones, ResNet-18 (He et al., 2016) and WideResNet (Zagoruyko & Komodakis, 2016), we test our method on: *mini*Imagenet and *tiered*ImageNet (Table 2). We follow the standard protocols (details of experimental settings in Appendix D). Our NIW-Meta exhibits consistent improvement over the SOTAs for different settings in support set size and backbones.

**Large-scale ViT backbones.** We also test our method on large-scale (pretrained) ViT backbones DINO-small (Dino/s) and DINO-base (DINO/b) (Caron et al., 2021), similarly as the setup in (Hu et al., 2022). In Table 3 we report results on the three benchmarks: *mini*Imagenet, CIFAR-FS, and *tiered*ImageNet. Our NIW-Meta adopts the same NCC head as ProtoNet after the ViT feature extractor. As claimed in (Hu et al., 2022), using the pretrained feature extractor and further finetuning it significantly boost the performance of few-shot learning algorithms including ours. Among the competing methods, our approach yields the best accuracy for most cases. In particular, compared to the shallow Bayesian MetaQDA (Zhang et al., 2021), treating all network weights as random variates in our model turns out to be more effective than the readout parameters alone.

Table 3: Classification accuracies with large-scale ViT backbones.

| Model | Backbone / Pretrain | *mini*ImageNet | | CIFAR-FS | | *tiered*ImageNet | |
|---|---|---|---|---|---|---|---|
| | | 1-shot | 5-shot | 1-shot | 5-shot | 1-shot | 5-shot |
| ProtoNet | DINO/s | $93.1^{\pm0.12}$ | $98.0^{\pm0.14}$ | $81.1^{\pm0.29}$ | $92.5^{\pm0.13}$ | $89.0^{\pm0.11}$ | $95.8^{\pm0.09}$ |
| MetaOpt | DINO/s | $92.2^{\pm0.22}$ | $97.8^{\pm0.16}$ | $70.2^{\pm0.22}$ | $84.1^{\pm0.27}$ | $87.5^{\pm0.25}$ | $94.7^{\pm0.20}$ |
| MetaQDA | DINO/s | $92.0^{\pm0.31}$ | $97.0^{\pm0.18}$ | $77.2^{\pm0.34}$ | $90.1^{\pm0.18}$ | $87.8^{\pm0.27}$ | $95.6^{\pm0.16}$ |
| **NIW-Meta** | DINO/s | $\mathbf{93.4^{\pm0.17}}$ | $\mathbf{98.2^{\pm0.15}}$ | $\mathbf{82.8^{\pm0.26}}$ | $\mathbf{92.9^{\pm0.11}}$ | $\mathbf{89.3^{\pm0.16}}$ | $\mathbf{96.0^{\pm0.14}}$ |
| ProtoNet | DINO/b | $95.3^{\pm0.13}$ | $98.4^{\pm0.12}$ | $84.3^{\pm0.19}$ | $92.2^{\pm0.13}$ | $91.2^{\pm0.15}$ | $96.5^{\pm0.10}$ |
| MetaOpt | DINO/b | $94.4^{\pm0.19}$ | $98.4^{\pm0.16}$ | $72.0^{\pm0.29}$ | $86.2^{\pm0.18}$ | $89.5^{\pm0.27}$ | $95.7^{\pm0.15}$ |
| MetaQDA | DINO/b | $94.7^{\pm0.21}$ | $\mathbf{98.7^{\pm0.14}}$ | $80.9^{\pm0.31}$ | $\mathbf{93.8^{\pm0.15}}$ | $89.7^{\pm0.21}$ | $96.5^{\pm0.07}$ |
| **NIW-Meta** | DINO/b | $\mathbf{95.5^{\pm0.15}}$ | $\mathbf{98.7^{\pm0.12}}$ | $\mathbf{84.7^{\pm0.13}}$ | $93.2^{\pm0.17}$ | $\mathbf{91.4^{\pm0.21}}$ | $\mathbf{96.7^{\pm0.11}}$ |

**Set-based adaptation backbones.** We also conduct experiments using the set-based adaptation architecture called FEAT introduced in (Ye et al., 2020). The network is tailored for

Table 4: FEAT vs. our method. Classification accuracies.

| Model | *mini*ImageNet | | *tiered*ImageNet | |
|---|---|---|---|---|
| | 1-shot | 5-shot | 1-shot | 5-shot |
| FEAT | 66.78 | 82.05 | $70.80^{\pm0.23}$ | $84.79^{\pm0.16}$ |
| **NIW-Meta (Ours)** | $\mathbf{66.91^{\pm0.10}}$ | $\mathbf{82.28^{\pm0.15}}$ | $\mathbf{70.93^{\pm0.27}}$ | $\mathbf{85.20^{\pm0.19}}$ |

few-shot adaptation, namely $y^Q = G(x^Q, S; \theta)$ where the network $G$ takes the entire support set $S$ and query image $x^Q$ as input. Note that our NIW-Meta can incorporate any network architecture, even the set-based one like FEAT. As shown in Table 4, the Bayesian treatment leads to further improvement over (Ye et al., 2020) with this set-based architecture.

**Error calibration.** Bayesian models are known to be better calibrated than deterministic counterparts. We measure the *expected calibration errors* (ECE) (Guo et al., 2017) to judge how well the prediction accuracy and the prediction confidence are aligned – $ECE = \sum_{b=1}^{B} \frac{N_b}{N} |acc(b) - conf(b)|$ where we partition test instances into $B$ bins along the model's prediction confidence scores, and $conf(b)$, $acc(b)$ are the average confidence and accuracy for the $b$-th bin, respectively. The results on *mini*ImageNet

Table 5: ECEs on *mini*ImageNet. "ECE+TS" indicates extra tuning of the temperature hyperparameter.

| Model | Backbone | ECE | | ECE+TS | |
|---|---|---|---|---|---|
| | | 1-shot | 5-shot | 1-shot | 5-shot |
| Linear classifier | Conv-4 | 8.54 | 7.48 | 3.56 | 2.88 |
| SimpleShot | Conv-4 | 33.45 | 45.81 | 3.82 | 3.35 |
| MetaQDA-MAP | Conv-4 | 8.03 | 5.27 | 2.75 | 0.89 |
| MetaQDA-FB | Conv-4 | 4.32 | 2.92 | 2.33 | 0.45 |
| **NIW-Meta (Ours)** | Conv-4 | **2.68** | **1.88** | **1.47** | **0.32** |
| SimpleShot | WRN-28-10 | 39.56 | 55.68 | 4.05 | 1.80 |
| S2M2+Linear | WRN-28-10 | 33.23 | 36.84 | 4.93 | 2.31 |
| MetaQDA-MAP | WRN-28-10 | 31.17 | 17.37 | 3.94 | 0.94 |
| MetaQDA-FB | WRN-28-10 | 30.68 | 15.86 | 2.71 | 0.74 |
| **NIW-Meta (Ours)** | WRN-28-10 | **10.79** | **7.11** | **2.03** | **0.65** |

are shown in Table 5. We used 20 bins and optionally performed the temperature search on validation sets, similarly as (Zhang et al., 2021). Again, Bayesian inference of whole network weights in our NIW-Meta leads to a far better calibrated model than the shallow Meta-QDA (Zhang et al., 2021).

## 7.2 FEW-SHOT REGRESSION

**Sine-Line dataset (Finn et al., 2018).** The $1D$ $(x, y)$ data pairs are generated by randomly selecting either linear or sine curves with different scales/slopes/frequencies/phases. For the episodic few-shot learning setup, we follow the standard protocol: each episode is comprised of $k = 5$-shot support and 45 query samples randomly drawn from a random curve (regarded as a task). To deal with real-valued targets, we adopt the so-called **RidgeNet**, which has a parameter-free readout head derived from the support data via (closed-form) estimation of the linear coefficient matrix using the ridge regression (the L2 regularisation coefficient $\lambda = 0.1$).

Table 6: Sine-Line results. PMAML w/ 5 inner steps incurred numerical errors.

| Model | Mean squared error | R-ECE |
|---|---|---|
| RidgeNet | 0.8210 | N/A |
| MAML (1-step) | 0.8206 | N/A |
| MAML (5-step) | 0.8309 | N/A |
| PMAML (1-step) | 0.9160 | 0.2666 |
| **NIW-Meta (Ours)** | **0.7822** | **0.1728** |

It is analogous to the ProtoNet (Snell et al., 2017) in classification which has a parameter-free head derived from NCC on support data. A similar model was introduced in (Bertinetto et al., 2019) but mainly repurposed for classification. We find that RidgeNet leads to much more accurate prediction than the conventional trainable linear head. For instance, the test errors are: RidgeNet = 0.82 vs. MAML with linear head = 1.86. Furthermore, we adopt the ridge head in other models as well, such as MAML, PMAML (Finn et al., 2018), and our NIW-Meta. See Table 6 for the mean squared errors contrasting our NIW-Meta against competing meta learning methods. The table also contains the regression-ECE (R-ECE) calibration errors[2]. Clearly our model is calibrated the best.

**Object pose estimation on ShapeNet datasets.** We consider the recent few-shot regression benchmarks (Gao et al., 2022; Yin et al., 2020) which introduced four datasets for object pose estimation: *Pascal-1D*, *ShapeNet-1D*, *ShapeNet-2D* and *Distractor*. In all datasets[3], the main goal is to estimate the pose (positions in pixel and/or rotation angles) of the target object in an image. Each episode is specified by: i) selecting a target object randomly from a pool of objects with different object categories, and ii) rendering the same object in an image with several different random poses (position/rotation) to generate data instances. There are $k$ support samples (input images and target pose labels) and $k_q$ query samples for each episode. For ShapeNet-1D, for instance, $k$ is randomly chosen from 3 to 15 while $k_q = 15$. Except Pascal-1D, some object categories are dedicated solely for

---

[2]The definition of the R-ECE is quite different from that of the classification ECE in Sec. 7.1. We follow the notion of *goodness of cumulative distribution matching* used in (Tran et al., 2020; Cui et al., 2020). Specifically, denoting by $\hat{Q}_p(x)$ the $p$-th quantile of the predicted distribution $\hat{p}(y|x)$, we measure the deviation of $p_{true}(y \leq \hat{Q}_p(x)|x)$ from $p$ by absolute difference. So it is 0 for the ideal case $\hat{p}(y|x) = p_{true}(y|x)$. Note that by definition we can only measure R-ECE for models with *probabilistic* output $\hat{p}(y|x)$.

[3]Pascal-1D and ShapeNet-1D are relatively easier datasets than the rest two as we have uniform noise-free backgrounds. To make the few-shot learning problem more challenging, ShapeNet-2D and Distractor datasets further introduce random (real-world) background images and/or so called the *distractors* which are objects randomly drawn and rendered that have nothing to do with the target pose to estimate.

Table 7: Pose estimation results. Mean squared errors in rotation angle differences (Pascal-1D and ShapeNet-1D), quaternion differences $\times 10^{-2}$ (ShapeNet-2D) and pixel errors (Distractor). The dataset-wise different augmentation schemes are shown in the parentheses.

| Model | Pascal-1D (TA) | ShapeNet-1D (TA+DA) | | ShapeNet-2D (TA+DA+DR) | | Distractor (DA) | |
|---|---|---|---|---|---|---|---|
| | | Intra-category | Cross-category | Intra-category | Cross-category | Intra-category | Cross-category |
| MAML | $1.02 \pm 0.06$ | 17.96 | 18.79 | – | – | – | – |
| CNP (Garnelo et al., 2018) | $1.98 \pm 0.22$ | $7.66 \pm 0.18$ | $8.66 \pm 0.19$ | $14.20^{\pm 0.06}$ | $13.56^{\pm 0.28}$ | 2.45 | 3.75 |
| CNP+BA (Volpp et al., 2021) | – | – | – | $14.16^{\pm 0.08}$ | $13.56^{\pm 0.18}$ | 2.44 | 3.97 |
| CNP+FCL (Gao et al., 2022) | – | – | – | – | – | 2.00 | 3.05 |
| ANP (Kim et al., 2019) | $1.36 \pm 0.25$ | $5.81 \pm 0.23$ | $6.23 \pm 0.12$ | $14.12^{\pm 0.14}$ | $13.59^{\pm 0.10}$ | 2.65 | 4.08 |
| ANP+FCL (Gao et al., 2022) | – | – | – | $14.01^{\pm 0.09}$ | $13.32^{\pm 0.18}$ | – | – |
| NIW-Meta w/ C+R | $\mathbf{0.89 \pm 0.06}$ | $5.62 \pm 0.38$ | $6.57 \pm 0.39$ | $21.25^{\pm 0.76}$ | $20.82^{\pm 0.43}$ | $8.90^{\pm 0.26}$ | $17.31^{\pm 0.38}$ |
| NIW-Meta w/ CNP | $0.94 \pm 0.15$ | $5.74 \pm 0.17$ | $6.91 \pm 0.18$ | $13.86^{\pm 0.20}$ | $13.04^{\pm 0.13}$ | $\mathbf{1.80^{\pm 0.01}}$ | $\mathbf{2.94^{\pm 0.14}}$ |
| NIW-Meta w/ ANP | $0.95 \pm 0.09$ | $\mathbf{5.47 \pm 0.12}$ | $\mathbf{6.06 \pm 0.18}$ | $13.74^{\pm 0.30}$ | $12.95^{\pm 0.48}$ | $3.10^{\pm 0.48}$ | $5.20^{\pm 0.88}$ |

meta testing and not revealed during training, thus yielding two different test scenarios: *intra-category* and *cross-category*, in which the test object categories are seen and unseen, respectively.

In (Gao et al., 2022), they test different augmentation strategies in their baselines: conventional *data augmentation* on input images (denoted by DA), *task augmentation* (TA) (Rajendran et al., 2020) which adds random noise to the target labels to help reducing the memorisation issue (Yin et al., 2020), and *domain randomisation* (DR) (Tobin et al., 2017) which randomly generates background images during training. Among several possible combinations reported in (Gao et al., 2022), we follow the strategies that perform the best. For the target error metrics (e.g., position Euclidean distances in pixels for Distractor, rotation angle differences for ShapeNet-1D), we follow the metrics used in (Gao et al., 2022). For instance, the quaternion metric may sound reasonable in ShapeNet-2D due to the non-uniform, non-symmetric structures that reside in the target space (3D rotation angles).

The results are summarised in Table 7. In (Gao et al., 2022), they have shown that the set-based backbone networks, especially the Conditional Neural Process (CNP) (Garnelo et al., 2018) and Attentive Neural Process (ANP) (Kim et al., 2019) outperform the conventional architectures of the conv-net feature extractor with the linear head that are adapted by MAML (Finn et al., 2017) (except for the Pascal-1D case). Motivated by this, we adopt the same set-based CNP/ANP architectures within our NIW-Meta. In addition, we also test the ridge-head model with the conv-net feature extractor (denoted by **C+R**). Two additional competing models contrasted here are: the Bayesian context aggregation in CNP (CNP+BA) (Volpp et al., 2021) and the use of the functional contrastive learning loss as extra regularisation (FCL) (Gao et al., 2022).

For Pascal-1D and ShapeNet-1D, there is a dataset regime where MAML clearly outperforms (Pascal-1D) and underperforms (ShapeNet-1D) the CNP/ANP architectures. Very promisingly, our NIW-Meta consistently performs the best for both datasets, regardless of the choice of the architectures: not just CNP/ANP but also conv-net feature extractor + ridge head (C+R). For ShapeNet-2D and Distractor where MAML is not reported due to the known computational issues and poor performance, our NIW-Meta still exhibits the best test performance with CNP/ANP architectures. Unfortunately, the conv-net + ridge head (C+R) did not work well, and our conjecture is that the presence of heavy noise and distractors in the input data requires more sophisticated modeling of interaction/relation among the input instances, as is mainly aimed (and successfully done) by CNP/ANP.

**Computational complexity, running time and memory footprint.** We have analysed the computational complexity of our NIW-Meta compared to the simple feed-forward workflows (e.g., ProtoNet). Our method incurs only constant-factor overhead compared to the minimal-cost ProtoNet, as summarised in Table 8 in Appendix E. Also in Fig. 5 therein, we also report the memory footprints and running times of MAML and our NIW-Meta on real datasets, which show that NIW-Meta has far lower memory requirement than MAML. MAML suffers from heavy use of memory and thecomputational overhead of keeping track of a large computational graph for inner gradient descent steps. Our NIW-Meta has a much more efficient strategy of local episodic optimisation that is linked to global parameters, without storing the full optimisation trace. Please see all details in Appendix E.

## 8 Conclusion

We have proposed a new hierarchical Bayesian perspective to the episodic FSL problem. By having a higher-level task-agnostic random variate and episode-wise task-specific variables, we formulate a principled Bayesian inference view of the FSL problem with a large number of tasks (evidence). The effectiveness of our approach has been verified empirically in terms of both prediction accuracy and calibration, on a wide range of classification/regression tasks with complex backbones including ViT and set-based adaptation networks.

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

# Appendix

## Table of Contents

## A  PROOFS FOR GENERALISATION ERROR BOUNDS

We prove the two theorems Theorem 5.1 and Theorem 5.2 in the main paper that upper-bound the generalisation error of the model that is averaged over the learned posterior $q(\phi, \theta_{1:N})$. Without loss of generality we assume $|D_i| = n$ for all episodes $i$. We let $(q^*(\phi), \{q_i^*(\theta_i)\}_{i=1}^N)$ be the optimal solution of (6). In these theorems we often make the assumption of $N \to \infty$, that the number of training episodes tends to infinity, mainly for mathematical convenience. This assumption may not be true for practical situations, however, the theorems can still be applicable with *approximate* guarantees where we provide several justifications for this in Sec. F.

### A.1  PROOF FOR PAC-BAYES-$\lambda$ BOUND

First, Theorem 5.1, reiterated below as Theorem A.1, relates the generalisation error to the ultimate ELBO loss (6) that we minimised in our algorithm.

**Theorem A.1** (PAC-Bayes-$\lambda$ bound). *Let $R_i(\theta)$ be the generalisation error of model $\theta$ for the task $i$, more specifically, $R_i(\theta) = \mathbb{E}_{(x,y) \sim \mathcal{T}_i}[-\log p(y|x, \theta)]$ with the assumption of $[0, 1]$-bounded errors. As the number of training episodes $N \to \infty$, the following holds with probability at least $1 - \delta$ for arbitrary small $\delta > 0$:*

$$\mathbb{E}_{i \sim \mathcal{T}} \mathbb{E}_{q_i^*(\theta_i)}[R_i(\theta_i)] \ \leq \ \frac{2\epsilon^*}{n}, \tag{16}$$

*where $\epsilon^*$ is the optimal value of (6).*

*Proof.* We utilise the recent PAC-Bayes-$\lambda$ bound (Thiemann et al., 2017; Rivasplata et al., 2019), a variant of the traditional PAC-Bayes bounds (McAllester, 1999; Langford & Caruana, 2001; Seeger, 2002; Maurer, 2004). It states that for any $\lambda \in (0, 2)$, the following holds with probability at least $1 - \delta$:

$$\mathbb{E}_{q(\beta)}[R(\beta)] \leq \frac{1}{1 - \lambda/2} \mathbb{E}_{q(\beta)}[\hat{R}_m(\beta)] + \frac{1}{\lambda(1 - \lambda/2)} \frac{\mathrm{KL}(q(\beta)||p(\beta)) + \log(2\sqrt{m}/\delta)}{m}, \tag{17}$$

where $\beta$ represents all model parameters (random variables), $R(\beta)$ is the generalisation error/loss for a given model $\beta$, and $\hat{R}_m(\beta)$ is the empirical error/loss on the training data of size $m$. It holds for any data-*independent* (e.g., prior) distribution $p(\beta)$ and any distribution (possibly data-dependent, e.g., posterior) $q(\beta)$.

Now, with $N \to \infty$, we rewrite (6) in an equivalent form as follows:

$$\min_{L_0, \{L_i\}_{i=1}^{\infty}} Q(L_0, \{L_i\}_{i=1}^{\infty}) := \frac{1}{N}\bigg( \mathbb{E}_{q(\phi;L_0)\prod_i q_i(\theta_i;L_i)}\big[\textstyle\sum_i l_i(\theta_i)\big] + \tag{18}$$

$$\left. \mathrm{KL}\Big(q(\phi;L_0)\textstyle\prod_i q_i(\theta_i;L_i) \,\big|\big|\, p(\phi)\textstyle\prod_i p(\theta_i|\phi)\Big)\bigg)\right|_{N\to\infty}$$

Then we set $\beta := \{\phi, \theta_{1:N}\}$, $q(\beta) := q(\phi)\prod_i q_i(\theta_i)$, and $p(\beta) := p(\phi)\prod_i p(\theta_i|\phi)$. We also define the generalisation loss and the empirical loss as follows:

$$R(\beta) := \frac{1}{N}\sum_{i=1}^{N}\mathbb{E}_{(x,y)\sim\mathcal{T}_i}[-\log p(y|x,\theta)] = \frac{1}{N}\sum_{i=1}^{N}R_i(\theta) \tag{19}$$

$$\hat{R}_m(\beta) := \frac{1}{N}\sum_{i=1}^{N}\mathbb{E}_{(x,y)\sim D_i}[-\log p(y|x,\theta)] = \frac{1}{n}\frac{1}{N}\sum_{i=1}^{N} -\log p(D_i|\theta_i) = \frac{1}{n}\frac{1}{N}\sum_{i=1}^{N}l_i(\theta_i) \tag{20}$$

Note that the empirical data size $m = nN$ in our case. Plugging these into (17) with $\lambda = 1$ leads to:

$$\frac{1}{N}\sum_{i=1}^{N}\mathbb{E}_{q_i(\theta_i)}[R_i(\theta_i)] \leq$$

$$2\left(\frac{1}{n}\frac{1}{N}\textstyle\sum_{i=1}^{N}\mathbb{E}_{q_i(\theta_i)}[l_i(\theta_i)] + \frac{\mathrm{KL}\big(q(\phi)\prod_i q_i(\theta_i)\big|\big|p(\phi)\prod_i p(\theta_i|\phi)\big) + \log(2\sqrt{nN}/\delta)}{nN}\right) \tag{21}$$

Taking $N \to \infty$ in (21) makes i) the LHS become $\mathbb{E}_{i\sim\mathcal{T}}\mathbb{E}_{q_i(\theta_i)}[R_i(\theta_i)]$, ii) the complexity term $\frac{\log(2\sqrt{nN}/\delta)}{nN}$ in the RHS vanish, and iii) the RHS converge to $\frac{2}{n}Q(L_0, \{L_i\}_{i=1}^{\infty})$. That is,

$$\mathbb{E}_{i\sim\mathcal{T}}\mathbb{E}_{q_i(\theta_i)}[R_i(\theta_i)] \leq \frac{2}{n}Q(L_0, \{L_i\}_{i=1}^{\infty}). \tag{22}$$

Since (22) holds for *any* $q$, we take the minimiser $q^*$ of (6), which completes the proof. $\qquad\square$

## A.2 Proof for Regression Analysis Bound

Theorem 5.2, reiterated below as Theorem A.2 in a more detailed form, is based on the recent regression analysis techniques (Pati et al., 2018; Bai et al., 2020). Before we prove the theorem, we formally state some core assumptions and notations. Let $P^i(x,y)$ be the *true* data distribution for episode/task $i$ where $i = 1, \ldots, N$ and $N \to \infty$. We consider regression-based data modeling, assuming that the target $y$ is real vector-valued ($y \in \mathbb{R}^{S_y}$). Also it is assumed that there exists a *true regression function* $f^i : \mathbb{R}^{S_x} \to \mathbb{R}^{S_y}$ for each $i$, more formally $P^i(y|x) = \mathcal{N}(y; f^i(x), \sigma_\epsilon^2 I)$, where $\sigma_\epsilon^2$ is constant Gaussian output noise variance.

For easier analysis we assume that the backbone network is an MLP with $L$ width-$M$ hidden layers, and all activation functions $\sigma(\cdot)$ are Lipschitz continuous with 1. We consider the bounded parameter space, $\theta \in \Theta = \{\theta \in \mathbb{R}^G : ||\theta||_\infty \leq B\}$, where $G = \dim(\theta)$ and $B$ is the maximal norm bound. Then the prediction (regression) function $f_\theta : \mathbb{R}^{S_x} \to \mathbb{R}^{S_y}$ is induced from $\theta$ as: $P_\theta(y|x) = \mathcal{N}(y; f_\theta(x), \sigma_\epsilon^2 I)$, where the true noise variance is assumed to be known. The expressions $\mathbb{E}_\theta[\cdot]$ and $\mathbb{E}^i[\cdot]$ refer to the expectations with respect to model's $P_\theta$ and the true $P^i$, respectively. The generalisation error measure that we consider is the *expected squared Hellinger distance* between the true $P^i$ and the model $P_\theta$, more specifically,

$$d^2(P_\theta, P^i) = \mathbb{E}_{x\sim P^i(x)}\big[H^2(P_\theta(y|x), P^i(y|x))\big] = \mathbb{E}_{x\sim P^i(x)}\left[1 - \exp\left(-\frac{||f_\theta(x) - f^i(x)||_2^2}{8\sigma_\epsilon^2}\right)\right]. \tag{23}$$

Now we state our theorem.

**Theorem A.2** (Bound derived from regression analysis). *Let $d^2(P_{\theta_i}, P^i)$ be the expected squared Hellinger distance between the true distribution $P^i(y|x)$ and model's $P_{\theta_i}(y|x)$ for task/episode $i$. Then the following holds with high probability:*

$$\mathbb{E}_{i\sim\mathcal{T}}\mathbb{E}_{q_i^*(\theta_i)}[d^2(P_{\theta_i}, P^i)] \leq \frac{C_0}{n} + C_1\epsilon_n^2 + C_2(r_n + \lambda^*), \tag{24}$$

where $C_\bullet > 0$ are some constant, $\lambda^* = \mathbb{E}_{i \sim \mathcal{T}}[\lambda_i^*]$ with $\lambda_i^* = \min_{\theta \in \Theta} \max_x ||\mathbb{E}_\theta[y|x] - \mathbb{E}^i[y|x]||^2$ is the lowest possible regression error within the underlying network $\Theta$, $r_n = \frac{G}{n}\left((L+1)\log M + \log\left(S_x\sqrt{\frac{n}{G}}\right)\right)$, and $\epsilon_n = \sqrt{r_n}\log^\delta(n)$ for $\delta > 1$ constant.

*Proof.* We utilise the Donsker-Varadhan's (DV) theorem (Boucheron et al., 2013) to relate the variational ELBO objective function to the Hellinger distance. The DV theorem says that the following inequality holds for any distributions $p$, $q$ and any (bounded) function $h(z)$:

$$\log \mathbb{E}_{p(z)}[e^{h(z)}] = \max_q \left(\mathbb{E}_{q(z)}[h(z)] - \mathrm{KL}(q||p)\right). \tag{25}$$

In our case, we define: $p(z) := p(\theta_i|\phi)$, $q(z) := q_i(\theta_i)$, $h(z) := \log \eta_i(\theta_i)$ with

$$\eta_i(\theta_i) := \exp\left(\rho(P_{\theta_i}(D_i), P^i(D_i)) + nd^2(P_{\theta_i}, P^i)\right) \tag{26}$$

where $\rho(P_{\theta_i}(D_i), P^i(D_i)) := \log \frac{P_{\theta_i}(D_i)}{P^i(D_i)}$ is the log-ratio. Note that $P(D_i) = P(Y_i|X_i)$. Plugging these into (25) leads to the following inequality which holds for any $\phi$:

$$n \cdot \mathbb{E}_{q_i(\theta_i)}[d^2(P_{\theta_i}, P^i)] \leq \mathbb{E}_{q_i(\theta_i)}[-\rho(P_{\theta_i}(D_i), P^i(D_i))] + \mathrm{KL}(q_i(\theta_i)||p(\theta_i|\phi)) + \log \mathbb{E}_{p(\theta_i|\phi)}[\eta_i(\theta_i)]. \tag{27}$$

We take the expectation with respect to $q(\phi)$, which yields:

$$n \cdot \mathbb{E}_{q_i(\theta_i)}[d^2(P_{\theta_i}, P^i)] \leq \mathbb{E}_{q_i(\theta_i)}[-\rho(P_{\theta_i}(D_i), P^i(D_i))] + \mathbb{E}_{q(\phi)}[\mathrm{KL}(q_i(\theta_i)||p(\theta_i|\phi))] + \mathbb{E}_{q(\phi)}\left[\log \mathbb{E}_{p(\theta_i|\phi)}[\eta_i(\theta_i)]\right]. \tag{28}$$

From the regression theorem (Pati et al., 2018) (Theorem 3.1 therein), it is known that $\mathbb{E}_{s(\theta)}[\eta(\theta)] \leq e^{Cn\epsilon_n^2}$ for any distribution $s(\theta)$ with high probability. We apply this result to the last term of (28). Summing it over $i = 1, \ldots, N$ leads to:

$$n \cdot \sum_{i=1}^N \mathbb{E}_{q_i(\theta_i)}[d^2(P_{\theta_i}, P^i)] \leq \sum_{i=1}^N \mathbb{E}_{q_i(\theta_i)}[-\rho(P_{\theta_i}(D_i), P^i(D_i))] + \sum_{i=1}^N \mathbb{E}_{q(\phi)}[\mathrm{KL}(q_i(\theta_i)||p(\theta_i|\phi))] + NCn\epsilon_n^2. \tag{29}$$

By dividing both sides by $N$ and sending $N \to \infty$, we have:

$$n \cdot \mathbb{E}_{i \sim \mathcal{T}}\mathbb{E}_{q_i(\theta_i)}[d^2(P_{\theta_i}, P^i)] \leq \underbrace{\mathbb{E}_{i \sim \mathcal{T}}\left[\mathbb{E}_{q_i(\theta_i)}[-\rho(P_{\theta_i}(D_i), P^i(D_i))] + \mathbb{E}_{q(\phi)}[\mathrm{KL}(q_i(\theta_i)||p(\theta_i|\phi))]\right]}_{= -\mathrm{ELBO}(q) + \log P^i(D_i)} + Cn\epsilon_n^2. \tag{30}$$

As indicated, the right hand side is composed of $-\mathrm{ELBO}(q)$ (the objective function of (6)), the constant $\log P^i(D_i)$, and the complexity term $Cn\epsilon_n^2$.

The next step is to plug in the optimal $q^*$ to have a meaningful upper bound. To this end, we introduce/define $\tilde{q}_i(\theta_i)$ and $\tilde{q}(\phi)$ as follows:

$$\tilde{q}_i(\theta_i) = \mathcal{N}(\theta_i; \theta_i^*, \sigma_n^2 I), \quad \tilde{q}(\phi) = \arg\min_{q(\phi)} \mathbb{E}_{i \sim \mathcal{T}}\mathbb{E}_{q(\phi)}[\mathrm{KL}(\tilde{q}_i(\theta_i)||p(\theta_i|\phi))], \quad \text{where} \tag{31}$$

$$\theta_i^* = \arg\min_{\theta \in \Theta} \max_{x \in \mathbb{R}^{S_x}} ||f_\theta(x) - f^i(x)||^2, \quad \sigma_n^2 = \frac{G}{8n}A, \tag{32}$$

$$A^{-1} = \log(3S_x M) \cdot (2BM)^{2(L+1)} \cdot \left(\left(S_x + 1 + \frac{1}{BM-1}\right)^2 + \frac{1}{(2BM)^2-1} + \frac{2}{(2BM-1)^2}\right). \tag{33}$$

Since $(\{q_i^*(\theta_i)\}_{i=1}^N, q^*(\phi))$ is the minimiser of the negative ELBO (6), we clearly have $-\text{ELBO}(q^*) \leq -\text{ELBO}(\tilde{q})$. We plug $q^*$ into (30) and apply this ELBO inequality to have:

$$
n \cdot \mathbb{E}_{i \sim \mathcal{T}} \mathbb{E}_{q_i^*(\theta_i)}[d^2(P_{\theta_i}, P^i)] \leq \mathbb{E}_{i \sim \mathcal{T}} \mathbb{E}_{\tilde{q}_i(\theta_i)}[-\rho(P_{\theta_i}(D_i), P^i(D_i))] +
$$
$$
\mathbb{E}_{i \sim \mathcal{T}} \mathbb{E}_{\tilde{q}(\phi)}[\text{KL}(\tilde{q}_i(\theta_i)||p(\theta_i|\phi))] + Cn\epsilon_n^2. \tag{34}
$$

The second term of the right hand side of (34) is constant (independent of $n$) and denoted by $\tilde{C}$. For the first term of the right hand side, we use the following fact from the proof of Lemma 4.1 in (Bai et al., 2020), which says that with high probability,

$$
\mathbb{E}_{\tilde{q}_i(\theta_i)}[-\rho(P_{\theta_i}(D_i), P^i(D_i))] \leq C'n(r_n + \lambda_i^*), \tag{35}
$$

for some constant $C' > 0$. Using this bound, (34) can be written as follows:

$$
n \cdot \mathbb{E}_{i \sim \mathcal{T}} \mathbb{E}_{q_i^*(\theta_i)}[d^2(P_{\theta_i}, P^i)] \leq \tilde{C} + C'n\left(r_n + \mathbb{E}_{i \sim \mathcal{T}}[\lambda_i^*]\right) + Cn\epsilon_n^2. \tag{36}
$$

The proof completes by dividing both sides by $n$. $\qquad\square$

## B DETAILED DERIVATIONS

### B.1 ELBO DERIVATION FOR (5)

We derive the upper bound of the negative marginal log-likelihood for our Bayesian FSL model, that is, deriving (5) in the main paper.

$$
\text{KL}\big(q(\phi, \theta_{1:N}) \,||\, p(\phi, \theta_{1:N}|D_{1:N})\big) = \mathbb{E}_q\left[\log \frac{q(\phi) \cdot \prod_i q_i(\theta_i) \cdot p(D_{1:N})}{p(\phi) \cdot \prod_i p(\theta_i|\phi) \cdot \prod_i p(D_i|\theta_i)}\right] \tag{37}
$$
$$
= \log p(D_{1:N}) +
$$
$$
\underbrace{\text{KL}(q(\phi)||p(\phi)) + \sum_{i=1}^N \left(\mathbb{E}_{q_i(\theta_i)}[-\log p(D_i|\theta_i)] + \mathbb{E}_{q(\phi)}\big[\text{KL}(q_i(\theta_i)||p(\theta_i|\phi))\big]\right)}_{=:\mathcal{L}(L)}. \tag{38}
$$

Since KL divergence is non-negative, $-\mathcal{L}(L)$ must be lower bound of the data log-likelihood $\log p(D_{1:N})$, rendering $\mathcal{L}(L)$ an upper bound of $-\log p(D_{1:N})$.

### B.2 DERIVATION FOR $\mathbb{E}_{q(\phi)}\big[\text{KL}(q_i(\theta_i)||p(\theta_i|\phi))\big]$ IN (6–7)

We will derive the full closed-form formula for $\mathbb{E}_{q(\phi)}\big[\text{KL}(q_i(\theta_i)||p(\theta_i|\phi))\big]$, which not only leads to equivalence between (7) and (8), but is also used in deriving (11). In a nutshell, the formula that we will prove is as follows:

$$
\mathbb{E}_{q(\phi)}\big[\text{KL}(q_i(\theta_i)||p(\theta_i|\phi))\big] = \tag{39}
$$
$$
\frac{1}{2}\left(-d\log(2e) + \log\frac{|V_0|}{|V_i|} - \psi_d\left(\frac{n_0}{2}\right) + \frac{d}{l_0} + n_0(m_i - m_0)^\top V_0^{-1}(m_i - m_0) + n_0\text{Tr}\big(V_i V_0^{-1}\big)\right),
$$

where $\psi_d(a) = \sum_{j=1}^d \psi(a + (1-j)/2)$ is the multivariate digamma function, and $\psi(\cdot)$ is the digamma function.

We begin with the definition of the KL divergence,

$$
\mathbb{E}_{q(\phi)}\big[\text{KL}(q_i(\theta_i)||p(\theta_i|\phi))\big] = -\mathbb{H}(q_i(\theta_i)) + \mathbb{E}_{q(\phi)q_i(\theta_i)}[-\log p(\theta_i|\phi)], \tag{40}
$$

where the first term is the negative entropy which admits a closed form due to Gaussian $q_i(\theta_i) = \mathcal{N}(\theta_i; m_i, V_i)$,

$$
-\mathbb{H}(q_i(\theta_i)) = -\frac{d}{2}\log(2\pi e) - \frac{1}{2}\log|V_i|. \tag{41}
$$

Next we expand the second term of (40) using $p(\theta_i|\phi) = \mathcal{N}(\theta_i; \mu, \Sigma)$ as follows:

$$\mathbb{E}_{q(\phi)q_i(\theta_i)}[-\log p(\theta_i|\phi)] = \underbrace{\frac{1}{2}\mathbb{E}_{q(\phi)}\big[\log|\Sigma|\big]}_{=:T_1} + \underbrace{\frac{1}{2}\mathbb{E}_{q(\phi)q_i(\theta_i)}\big[(\theta_i - \mu)^\top \Sigma^{-1}(\theta_i - \mu)\big]}_{=:T_2} + \frac{d}{2}\log(2\pi).$$

$$\tag{42}$$

Using the following facts from (Bishop, 2006; Braun & McAuliffe, 2008):

$$\mathbb{E}_{\mathcal{IW}(\Sigma;\Psi,\nu)}\log|\Sigma| = -d\log 2 + \log|\Psi| - \psi_d(\nu/2) \tag{43}$$

$$\mathbb{E}_{\mathcal{IW}(\Sigma;\Psi,\nu)}\Sigma^{-1} = \nu\Psi^{-1}, \tag{44}$$

we can derive the two terms $T_1$ and $T_2$ as follows (Recall: $q(\phi) = \mathcal{N}(\mu; m_0, l_0^{-1}\Sigma) \cdot \mathcal{IW}(\Sigma; V_0, n_0)$):

$$(T_1 =)\ \frac{1}{2}\mathbb{E}_{q(\phi)}\big[\log|\Sigma|\big] = \frac{1}{2}\Big(-d\log 2 + \log|V_0| - \psi_d\Big(\frac{n_0}{2}\Big)\Big) \tag{45}$$

$$(T_2 =)\ \frac{1}{2}\mathbb{E}_{q(\phi)q_i(\theta_i)}\big[(\theta_i - \mu)^\top \Sigma^{-1}(\theta_i - \mu)\big] = \frac{1}{2}\mathbb{E}_{q(\phi)q_i(\theta_i)}\mathrm{Tr}\Big((\theta_i - \mu)(\theta_i - \mu)^\top \Sigma^{-1}\Big) \tag{46}$$

$$= \frac{1}{2}\mathrm{Tr}\Big(\mathbb{E}_{q(\phi)}\big[\mathbb{E}_{q_i(\theta_i)}[(\theta_i - \mu)(\theta_i - \mu)^\top]\Sigma^{-1}\big]\Big) \tag{47}$$

$$= \frac{1}{2}\mathrm{Tr}\Big(\mathbb{E}_{q(\phi)}\big[(m_i m_i^\top - \mu m_i^\top - m_i\mu^\top + \mu\mu^\top + V_i)\Sigma^{-1}\big]\Big) \tag{48}$$

$$= \frac{1}{2}\mathrm{Tr}\Big(\mathbb{E}_{\mathcal{IW}(\Sigma;V_0,n_0)}\big[\mathbb{E}_{\mathcal{N}(\mu;m_0,l_0^{-1}\Sigma)}[m_i m_i^\top - \mu m_i^\top - m_i\mu^\top + \mu\mu^\top + V_i]\Sigma^{-1}\big]\Big) \tag{49}$$

$$= \frac{1}{2}\mathrm{Tr}\Big(\mathbb{E}_{\mathcal{IW}(\Sigma;V_0,n_0)}\big[(m_i m_i^\top - m_0 m_i^\top - m_i m_0^\top + m_0 m_0^\top + l_0^{-1}\Sigma + V_i)\Sigma^{-1}\big]\Big) \tag{50}$$

$$= \frac{1}{2}\mathrm{Tr}\Big(\frac{1}{l_0}I + \big((m_i - m_0)(m_i - m_0)^\top + V_i\big)n_0 V_0^{-1}\Big) \tag{51}$$

$$= \frac{1}{2}\Big(\frac{d}{l_0} + n_0(m_i - m_0)^\top V_0^{-1}(m_i - m_0) + n_0\mathrm{Tr}\big(V_i V_0^{-1}\big)\Big) \tag{52}$$

Combining all the above results yields the formula (39).

### B.3 DERIVATION FOR (8) FROM (7)

Using the result (39), we can easily show that the local episodic optimisation (7) in the main paper ((53) below) reduces to (8) ((54) below).

$$\min_{L_i}\ \mathbb{E}_{q_i(\theta_i;L_i)}[l_i(\theta_i)] + \mathbb{E}_{q(\phi)}\big[\mathrm{KL}(q_i(\theta_i; L_i)||p(\theta_i|\phi))\big] \tag{53}$$

$$\min_{m_i,V_i}\ \mathbb{E}_{\mathcal{N}(\theta_i;m_i,V_i)}[l_i(\theta_i)] - \frac{1}{2}\log|V_i| + \frac{n_0}{2}(m_i - m_0)^\top V_0^{-1}(m_i - m_0) + \frac{n_0}{2}\mathrm{Tr}\big(V_i V_0^{-1}\big) \tag{54}$$

Recall that the optimisation is with respect to $L_i = (m_i, V_i)$ with $L_0 = \{m_0, V_0, l_0, n_0\}$ fixed. Plugging (39) into (53) and removing the terms other than $(m_i, V_i)$ leads to (54).

### B.4 DERIVATION FOR (10)

For the quadratic approximation of $l_i(\theta_i) = -\log p(D_i|\theta_i) \approx \frac{1}{2}(\theta_i - \overline{m}_i)^\top \overline{A}_i(\theta_i - \overline{m}_i) + \text{const.}$, here we show that the minimiser of (8) ((54) above) can be obtained by the closed-form formula (10) ((55) below).

$$m_i^*(L_0) = (\overline{A}_i + n_0 V_0^{-1})^{-1}(\overline{A}_i\overline{m}_i + n_0 V_0^{-1}m_0), \qquad V_i^*(L_0) = (\overline{A}_i + n_0 V_0^{-1})^{-1}. \tag{55}$$

By replacing $l_i(\theta_i)$ by the quadratic approximation, the expected loss term in (8) or (54) can be written as follows:

$$\mathbb{E}_{\mathcal{N}(\theta_i;m_i,V_i)}[l_i(\theta_i)] \approx \mathbb{E}_{\mathcal{N}(\theta_i;m_i,V_i)}\Big[\frac{1}{2}(\theta_i - \overline{m}_i)^\top \overline{A}_i(\theta_i - \overline{m}_i)\Big] + \text{const.} \tag{56}$$

$$= \frac{1}{2}\Big(\mathrm{Tr}\big(\mathbb{E}[\theta\theta^\top]\overline{A}_i\big) - \overline{m}_i^\top \overline{A}_i m_i - m_i^\top \overline{A}_i\overline{m}_i + \overline{m}_i^\top \overline{A}_i\overline{m}_i\Big) + \text{const.} \tag{57}$$

---

**Algorithm 2** Meta-test prediction algorithm.

---

**Input:** Test support data $D^*$ and learned $q(\phi; L_0)$ where $L_0 = \{m_0, V_0, n_0\}$.
$\quad\quad M_V$ = number of test-time variational inference steps.
$\quad\quad M_S$ = number of test-time model samples.
Compute the mode $\phi^* = (\mu^* = m_0, \Sigma^* = V_0/(n_0+d+2))$.
Initialise $(m, V)$ with $(\mu^*, \Sigma^*)$.
**for** $i = 1, \ldots, M_V$ **do**
$\quad$ Take a gradient descent update for $(m, V)$ with the objective in (64).
**end for**
Sample $\theta^{(s)} \sim \mathcal{N}(\theta; m, V)$ for $s = 1, \ldots, M_S$.
**Output:** Sample-averaged predictive distribution, $p(y^*|x^*, D^*, D_{1:\infty}) \approx \frac{1}{S} \sum_{s=1}^{M_S} p(y^*|x^*, \theta^{(s)})$.

---

$$= \frac{1}{2}\Big(\mathrm{Tr}\big(V_i \overline{A}_i\big) + m_i^\top \overline{A}_i m_i - \overline{m}_i^\top \overline{A}_i m_i - m_i^\top \overline{A}_i \overline{m}_i + \overline{m}_i^\top \overline{A}_i \overline{m}_i\Big) + \mathrm{const.} \quad (58)$$

$$= \frac{1}{2}\Big(\mathrm{Tr}\big(V_i \overline{A}_i\big) + (m_i - \overline{m}_i)^\top \overline{A}_i(m_i - \overline{m}_i)\Big) + \mathrm{const.} \quad (59)$$

After plugging this back to (54), we take the derivatives of the objective with respect to $m_i$ and $V_i$ and set them to 0:

$$\nabla_{m_i}(\cdot) \;=\; \overline{A}_i(m_i - \overline{m}_i) + n_0 V_0^{-1}(m_i - m_0) \;=\; 0 \quad (60)$$

$$\nabla_{V_i}(\cdot) \;=\; \frac{1}{2}\Big(\overline{A}_i - V_i^{-1} + n_0 V_0^{-1}\Big) \;=\; 0 \quad (61)$$

The solution becomes (10) or (55).

## B.5 Derivation for (11)

It is quite straightforward that by plugging (10) or (55) and also (39) in (6), we have our final optimisation problem (11) in the main paper. It is reiterated below:

$$\min_{L_0} \; \mathbb{E}_{i \sim \mathcal{T}}\Big[f_i(L_0) + \frac{1}{2}g_i(L_0) + \frac{d}{2l_0}\Big] \quad \mathrm{s.t.} \quad f_i(L_0) \;=\; \mathbb{E}_{\epsilon \sim \mathcal{N}(0,I)}\Big[l_i\Big(m_i^*(L_0) + V_i^*(L_0)^{1/2}\epsilon\Big)\Big],$$

$$g_i(L_0) \;=\; \log \frac{|V_0|}{|V_i^*(L_0)|} + n_0 \mathrm{Tr}\big(V_i^*(L_0)/V_0\big) + n_0\big(m_i^*(L_0) - m_0\big)^2/V_0 - \psi_d\Big(\frac{n_0}{2}\Big), \quad (62)$$

## B.6 Formulas for Test-Time ELBO Optimisation (13)

We provide formulas for the test-time ELBO in (13) ((63) below). For the test-time variational density $v(\theta) = \mathcal{N}(\theta; m, V)$ to approximate $p(\theta|D^*, \phi^*)$ for test support data $D^*$ and learned $\phi^* = (\mu^* = m_0, \Sigma^* = V_0/(n_0+d+2))$, we had

$$\min_{m,V} \; \mathbb{E}_{v(\theta)}[-\log p(D^*|\theta)] + \mathrm{KL}(v(\theta)||p(\theta|\phi^*)). \quad (63)$$

Using the closed-form Gaussian KL divergence and the reparametrised sampling trick, we can express (63) as:

$$\min_{m,V} \; \Bigg\{ \mathbb{E}_{\epsilon \sim \mathcal{N}(0,I)}\big[-\log p\big(D^*|m + V^{1/2}\epsilon\big)\big] - \frac{1}{2}\log|V| +$$

$$\frac{n_0+d+2}{2}\Big(\mathrm{Tr}\big(V_0^{-1}V\big) + (m - m_0)^\top V_0^{-1}(m - m_0)\Big)\Bigg\}. \quad (64)$$

Also, our meta-test prediction algorithm is summarised as a pseudo code in Alg. 2.

# C Toy Experiment: Why Hierarchical Bayesian Model? (A Detailed Version)

To demonstrate why our hierarchical Bayesian modelling is effective for few-shot meta learning problems, we devise a simple toy synthetic experiment as a proof of concept.

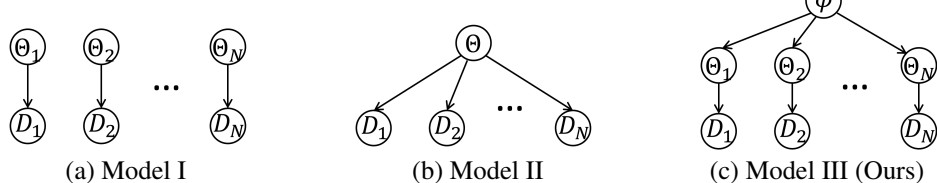

Figure 3: (Toy experiment) Graphical models for the three competing Bayesian models. Here $\Theta = [\theta, \beta]$ is the the concatenation of the weight and intercept random variables.

The problem that we consider is basically a multi-task (Bayesian) linear regression problem. First, we generate the multi-task/episodic data by the following process: The input-output data pairs $(x \in \mathbb{R}^2, y \in \mathbb{R})$ are generated from a linear model with a shared normal vector and episode-specific intercepts. More specifically, let $w_{\text{shared}} \in \mathbb{R}^2$ be the episode-agnostic shared weight vector, and $\{b_1, b_2, b_3\}$ $(b_j \in \mathbb{R})$ be the three candidate intercepts among which each episode can take one randomly. The actual values of the true parameters are: $w_{\text{shared}} = [-5.4282, 4.9867], b_1 = 1.4149, b_2 = -7.5315, b_3 = -2.8930$. The true data distribution $\mathcal{T}_i$ for the episode $i (= 1, 2, \ldots, N)$ is defined by the following linear process:

1. Sample the intercept ID for this episode, $j(i) \sim \{1, 2, 3\}$ uniformly at random.

2. Repeat the following to collect $(x, y)$ pairs (so that $(x, y) \sim \mathcal{T}_i$):

$$y = (w_{\text{shared}} + \epsilon_w)^\top x + b_{j(i)} + \epsilon_y, \tag{65}$$

where $x \sim \mathcal{N}(0, I)$, $\epsilon_w \sim \mathcal{N}(0, 10^{-4}I)$, and $\epsilon_y \sim \mathcal{N}(0, 10^{-4})$.

In this way we ensure that the resulting episodes are not only related to one another through the shared weight vector $w_{\text{shared}}$, but they are differentiated by potentially different intercepts. We generate 50 episodes where $N = 40$ episodes are used for training and the rest 10 episodes serve as test data. For each training episode $i$, we have three $(x, y)$ samples as an episodic training set $D_i$ (all available to a training algorithm, where we make no distinction between support and query sets). At test time, we take three samples as a (labeled) support set $D_*$ ($*$ denotes each of the 10 test episodes), and test performance is measured on about 50 unseen samples from the same distribution $\mathcal{T}_*$.

**Three competing Bayesian methods.** We consider three Bayesian models which exhibit different levels/degrees of flexibility and regularisation. The first one is highly flexible by modeling each individual episode independently with its own parameters, thus with lack of regularisation. The second case is a conventional (non-hierarchical) Bayesian model where we consider a single parameter set shared across episodes, thus too much regularised with lack of flexibility. At last, our hierarchical Bayesian model imposes balanced flexibility and regularisation by introducing the higher-level variable $\phi$ that captures the inter-episode shared information.

1. **Model I**: This model has episode-wise parameters while they are all independent with minimal regularisation. More formally,

$$y = \theta_i^\top x + \beta_i + \epsilon_y \tag{66}$$

where $(\theta_i, \beta_i)$ is the parameters for the episode $i$. We place the prior $p(\theta_i, \beta_i) = \mathcal{N}(\mu, 10^{-4}I)$ with the model parameter $\mu$ shared over episodes. The training amounts to learning the parameter $\mu \in \mathbb{R}^3$ via marginal likelihood maximisation (i.e., $\max_\mu \log p(D_1, \ldots, D_N | \mu)$). At test time we do inference $p(\theta_*, \beta_* | D_*, D_{1:N})$ which boils down to $p(\theta_*, \beta_* | D_*)$ due to the cross-episode independence assumption. The graphical model diagram for the model is shown in Fig. 3(a).

2. **Model II**: Unlike introducing episode-specific variables, this model has a single set of variables shared across all episodes. More specifically,

$$y = \theta^\top x + \beta + \epsilon_y \tag{67}$$

where $(\theta, \beta)$ is the episode-agnostic parameters, endowed with the prior $p(\theta, \beta) = \mathcal{N}(\mu, 10^{-4}I)$. Due to this parameter sharing, this model is highly regularised but at the expense of significantly reduced flexibility. Once $\mu$ is trained, the inference at test time is done by $p(\theta, \beta | D_*, D_{1:N})$ which is not simplified further and has to take into account all training and test data. The graphical model diagram in Fig. 3(b).

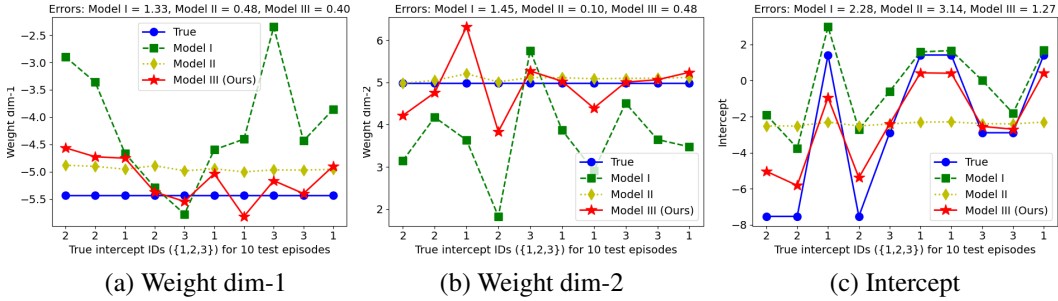

(a) Weight dim-1         (b) Weight dim-2         (c) Intercept

Figure 4: Toy experiments. Visualisation of the learned posterior means compared to the true values (blue-circled). (a) weight dim-1 ($\theta[0]$ vs. $w_{\text{shared}}[0]$), (b) weight dim-2 ($\theta[1]$ vs. $w_{\text{shared}}[1]$) and (c) intercept ($\beta$ vs. $b_{j(*)}$). In each plot, the X-axis shows the indices of the true intercepts sampled, that is, $j(*) \in \{1, 2, 3\}$, for 10 test episodes. In the titles we also report the distances (errors) between the true values and the posterior means for the three methods, averaged over 10 episodes.

3. **Model III (Ours)**: This is our hierarchical model where each episode has its own parameters (like Model I) but there is a globally governing variable $\phi$ to regularise the episode-wise parameters. That is, we have the same regression form as (66) with episode-wise $(\theta_i, \beta_i)$ parameters, but our prior distributions are defined hierarchically as:

$$p(\phi) = \mathcal{N}(m, V), \quad p(\theta_i, \beta_i|\phi) = \mathcal{N}(\phi, 10^{-4}I), \tag{68}$$

where $(m, V)$ is the model parameters to be learned. At test time, we infer $p(\theta_*, \beta_*|D_*, D_{1:N})$, however, unlike the Model II case, each training data $D_i$ and the test data $D_*$ do not have equal, symmetric contribution. Note that the training data affect the posterior indirectly only through the higher-level variable $\phi$ as follows:

$$p(\theta_*, \beta_*|D_*, D_{1:N}) = \int p(\theta_*, \beta_*|\phi, D_*) \, p(\phi|D_*, D_{1:N}) \, d\phi. \tag{69}$$

That is, while our model is as flexible as Model I due to the episode-wise parameters, their impacts on the test prediction are controlled/regularised in a very sensible manner. The graphical model diagram in Fig. 3(c).

Note that all the posterior inferences of the above three models can be done in *closed forms* due to the linear-Gaussian properties. The details of the posterior distributions and derivations as well as the model training can be found in Sec. C.1.

**Results.** For the 10 test episodes, we obtain the posterior means of the weights and intercept parameters $\mathbb{E}[\theta, \beta|D_*, D_{1:N}]$ for the three models via the closed-form solutions as detailed in Sec. C.1. The test support set size $|D_*| = 3$. With these posterior means, we predict the outputs of about 50 unseen test inputs. The mean absolute errors (MAE) of the three models averaged over 10 test episodes are: **Model I = 2.87**, **Model II = 3.13**, and **Model III (ours) = 1.28**, clearly showing the superiority of our model to other competing methods. In Fig. 4 we also visualise the posterior means to check how much they deviate from the true weights and intercept, namely the difference between the posterior mean of $(\theta, \beta)$ and the true $(w_{\text{shared}}, b_{j(*)})$. First, we see that **Model II**'s posterior means rarely change over the test episodes, in other words, the impact of test support data $D_*$ is diminished. This behavior is expected since the model imposes too much regularisation with little flexibility, and the test prediction is dominated by the mean model obtained from training data $D_{1:N}$. Secondly, **Model I** exhibits highly sensitive predictions over the test episodes, which mainly originates from little regularisation. In other words, the posterior is affected too sensitively by the current episode's support data, thus being vulnerable to overfitting especially when the support data size is small, typical in the few-shot learning. The model failed to capture useful shared information, in this case $w_{\text{shared}}$, from diverse training episodes. On the other hand, our **Model III** takes the balance between the above two extremes, imposing proper amount of regularisation and endowing adequate flexibility. Our posterior estimation best extracts the shared episode-agnostic information (the weight parameters fluctuate less over the test episodes), and at the same time, captures the episode-specific features the most accurately (the estimated intercepts are aligned well with the true values).

## C.1 DERIVATIONS FOR TRAINING AND POSTERIORS IN TOY EXPERIMENTS

For the convenience in notation, we let $\Theta = [\theta, \beta] \in \mathbb{R}^3$ be the concatenated random variables (subscripts can be applied accordingly). The training data $D_i$ for each episode $i$ consists of the *inputs*

with the constant 1 appended, denoted as $X_i \in \mathbb{R}^{n \times 3}$, and the outputs $Y_i \in \mathbb{R}^n$. With this notation, the linear regression model can be succinctly written as $Y_i = \Theta X_i$. Similarly, the test-time support data $D_*$ is decomposed into $(X_*, Y_*)$. The noise standard deviation is denoted by $\sigma = 10^{-2}$.

### C.1.1 MODEL I

Model I (the episode-wise independent model) can be formally written as:

$$p(\Theta_i) = \mathcal{N}(\Theta_i; \mu, \sigma^2 I), \quad p(D_i|\Theta_i) = \mathcal{N}(Y_i; X_i\Theta_i, \sigma^2 I), \quad \forall i. \tag{70}$$

The posterior $p(\Theta_1, \ldots, \Theta_N | D_1, \ldots, D_N)$ is fully factorised over $i$, and we can deal with individual terms $p(\Theta_i|D_i)$ where

$$p(\Theta_i|D_i) \propto p(\Theta_i) \cdot p(D_i|\Theta_i) = \mathcal{N}(\Theta_i; \mu, \sigma^2 I) \cdot \mathcal{N}(Y_i; X_i\Theta_i, \sigma^2 I). \tag{71}$$

Due to the product-of-Gaussians form, we have the closed-form posterior:

$$p(\Theta_i|D_i) = \mathcal{N}(\Theta_i; A_i^{-1} b_i, A_i^{-1}) \text{ where } A_i = \frac{1}{\sigma^2}(I + X_i^\top X_i), \; b_i = \frac{1}{\sigma^2}(\mu + X_i^\top Y_i). \tag{72}$$

The training amounts to maximising the data (log-)likelihood, $\max_\mu \log p(D_1, \ldots, D_N | \mu)$ where the objective is fully decomposed over $i$ as follows:

$$\log p(D_1, \ldots, D_N | \mu) = \sum_{i=1}^N \log p(D_i) \quad \text{where} \tag{73}$$

$$\log p(D_i) = \log p(\Theta_i, D_i) - \log p(\Theta_i|D_i) \quad \text{(for any } \Theta_i) \tag{74}$$

$$= \log p(\Theta_i) + \log p(D_i|\Theta_i) - \log p(\Theta_i|D_i) \quad \text{(for any } \Theta_i). \tag{75}$$

Using the Gaussian posterior form (72), we can easily evaluate $\log p(D_1, \ldots, D_N | \mu)$ and also optimise it with respect to $\mu$. At test time, the posterior of $\Theta_*$ given all the training data and the test support data, $p(\Theta_*|D_*, D_1, \ldots, D_N)$ equals $p(\Theta_*|D_*)$ due to the independence assumption, and admits the same Gaussian form as (72) with the test support data $(X_*, Y_*)$ in the place of $(X_i, Y_i)$.

### C.1.2 MODEL II

Model II (the shared model across episodes) can be formally written as:

$$p(\Theta) = \mathcal{N}(\Theta; \mu, \sigma^2 I), \quad p(D_i|\Theta) = \mathcal{N}(Y_i; X_i\Theta, \sigma^2 I). \tag{76}$$

The posterior $p(\Theta|D_1, \ldots, D_N)$ can be derived as follows:

$$p(\Theta|D_1, \ldots, D_N) \propto p(\Theta) \cdot \prod_{i=1}^N p(D_i|\Theta) = \mathcal{N}(\Theta; \mu, \sigma^2 I) \cdot \prod_{i=1}^N \mathcal{N}(Y_i; X_i\Theta, \sigma^2 I). \tag{77}$$

Again, due to the product-of-Gaussians form, we have the closed-form posterior:

$$p(\Theta|D_1, \ldots, D_N) = \mathcal{N}(\Theta; A^{-1}b, A^{-1}) \text{ where}$$

$$A = \frac{1}{\sigma^2}\left(I + \sum_{i=1}^N X_i^\top X_i\right), \; b_i = \frac{1}{\sigma^2}\left(\mu + \sum_{i=1}^N X_i^\top Y_i\right). \tag{78}$$

Likewise, the training amounts to maximising the data (log-)likelihood, $\max_\mu \log p(D_1, \ldots, D_N | \mu)$ where the objective becomes:

$$\log p(D_1, \ldots, D_N | \mu) = \log p(\Theta, D_1, \ldots, D_N) - \log p(\Theta|D_1, \ldots, D_N) \quad \text{(for any } \Theta) \tag{79}$$

$$= \log p(\Theta) + \sum_{i=1}^N \log p(D_i|\Theta) - \log p(\Theta|D_1, \ldots, D_N) \quad \text{(for any } \Theta). \tag{80}$$

Again, using the Gaussian posterior form (78), we can easily evaluate $\log p(D_1, \ldots, D_N | \mu)$ and also optimise it with respect to $\mu$. At test time, the posterior of $\Theta_*$ given all the training data and the test support data, $p(\Theta_*|D_*, D_1, \ldots, D_N)$ admits a Gaussian form, derived similarly as (78) with the test support data statistics $X_*^\top X_*$ and $X_*^\top Y_*$ additionally added to the training statistics.

### C.1.3 MODEL III (OURS)

Our Model III (the hierarchical Bayesian model) can be formally written as:

$$p(\phi) = \mathcal{N}(\phi; m, V), \tag{81}$$

$$p(\Theta_i|\phi) = \mathcal{N}(\Theta_i; \phi, \sigma^2 I), \quad p(D_i|\Theta_i) = \mathcal{N}(Y_i; X_i\Theta_i, \sigma^2 I), \quad \forall i. \tag{82}$$

The posterior $p(\phi|D_1, \ldots, D_N)$ can be derived as follows:

$$p(\phi|D_1, \ldots, D_N) \propto p(\phi) \cdot \int p(\Theta_1, \ldots, \Theta_N|\phi) \cdot p(D_1, \ldots, D_N|\Theta_1, \ldots, \Theta_N) \, d\Theta_{1:N} \tag{83}$$

$$= p(\phi) \cdot \int \prod_{i=1}^{N} \left( p(\Theta_i|\phi) \cdot p(D_i|\Theta_i) \right) d\Theta_{1:N} \tag{84}$$

$$= p(\phi) \cdot \prod_{i=1}^{N} \int p(\Theta_i|\phi) \cdot p(D_i|\Theta_i) \, d\Theta_i \tag{85}$$

$$= \mathcal{N}(\phi; m, V) \cdot \prod_{i=1}^{N} \int \mathcal{N}(\Theta_i; \phi, \sigma^2 I) \cdot \mathcal{N}(Y_i; X_i\Theta_i, \sigma^2 I) \, d\Theta_i \tag{86}$$

$$= \mathcal{N}(\phi; m, V) \cdot \prod_{i=1}^{N} \mathcal{N}(Y_i; X_i\phi, \sigma^2(I + X_i X_i^\top)), \tag{87}$$

where we use the property of the product of Gaussians for the derivation from (86) to (87). Now, in (87), due to the product-of-Gaussians form, we have the closed-form posterior:

$$p(\phi|D_1, \ldots, D_N) = \mathcal{N}(\phi; A^{-1}b, A^{-1}) \quad \text{where}$$

$$A = V^{-1} + \frac{1}{\sigma^2} \sum_{i=1}^{N} X_i^\top (X_i X_i^\top + I)^{-1} X_i, \ b = V^{-1}m + \frac{1}{\sigma^2} \sum_{i=1}^{N} X_i^\top (X_i X_i^\top + I)^{-1} Y_i. \tag{88}$$

The training amounts to maximising the data (log-)likelihood, $\max_{m,V} \log p(D_1, \ldots, D_N|m, V)$ where the objective becomes:

$$\log p(D_1, \ldots, D_N|m, V) = \log p(\phi, D_1, \ldots, D_N) - \log p(\phi|D_1, \ldots, D_N) \quad \text{(for any } \phi) \tag{89}$$

$$= \log p(\phi) + \log p(D_1, \ldots, D_N|\phi) - \log p(\phi|D_1, \ldots, D_N) \quad \text{(for any } \phi), \tag{90}$$

and the $\phi$-conditioned data log-likelihood $\log p(D_1, \ldots, D_N|\phi)$ can be derived as follows:

$$\log p(D_1, \ldots, D_N|\phi) = \log \int \prod_{i=1}^{N} \left( p(\Theta_i|\phi) \cdot p(D_i|\Theta_i) \right) d\Theta_{1:N} \tag{91}$$

$$= \log \prod_{i=1}^{N} \int p(\Theta_i|\phi) \cdot p(D_i|\Theta_i) \, d\Theta_i \tag{92}$$

$$= \sum_{i=1}^{N} \log \int p(\Theta_i|\phi) \cdot p(D_i|\Theta_i) \, d\Theta_i \tag{93}$$

$$= \sum_{i=1}^{N} \log \mathcal{N}(Y_i; X_i\phi, \sigma^2(I + X_i X_i^\top)). \tag{94}$$

So we can easily evaluate $\log p(D_1, \ldots, D_N|m, V)$ and also optimise it with respect to $(m, V)$. At test time, the posterior of $\Theta_*$ given all the training data and the test support data, $p(\Theta_*|D_*, D_1, \ldots, D_N)$ can be derived as follows. We start with:

$$p(\Theta_*|D_*, D_1, \ldots, D_N) = \int p(\Theta_*|\phi, D_*) \cdot p(\phi|D_*, D_1, \ldots, D_N) \, d\phi, \tag{95}$$

and the first term in the integration, $p(\Theta_*|\phi, D_*)$, since it is proportional to the product of two Gaussians $p(\Theta_*|\phi, D_*) \propto p(D_*|\Theta_*) \cdot p(\Theta_*|\phi) = \mathcal{N}(Y_*; X_*\Theta_*, \sigma^2 I) \cdot \mathcal{N}(\Theta_*; \phi, \sigma^2 I)$, it admits Gaussian (from the Gaussian properties),

$$p(\Theta_*|\phi, D_*) = \mathcal{N}(\Theta_*; C\phi + d, E), \quad \text{where} \tag{96}$$

$$C = I - KX_*, \;\; d = KY_*, \;\; E = \sigma^2 C, \;\; K = X_*^\top (X_* X_*^\top + I)^{-1}. \tag{97}$$

The second term of (95) becomes a Gaussian following the derivation similar to (88) with the test support data $X_*$ and $Y_*$ included. Consequently we let $p(\phi|D_*, D_1, \ldots, D_N) = \mathcal{N}(\phi; A_*^{-1} b_*, A_*^{-1})$. At last, (95) is the marginalisation of the product of two Gaussians, which admits the following closed form:

$$p(\Theta_*|D_*, D_1, \ldots, D_N) = \mathcal{N}(\Theta_*; CA_*^{-1} b_* + d, CA_*^{-1} C^\top + E). \tag{98}$$

## D  IMPLEMENTATION DETAILS AND EXPERIMENTAL SETTINGS

We implement our NIW-Meta using PyTorch (Paszke et al., 2017) and the Higher (Grefenstette et al., 2019)[4] library. The latter makes the implementation of the backpropagation through the functional network weights in PyTorch modules very easy. Real codes for the synthetic SineLine regression dataset and the large-scale ViT are also attached in the Supplement Material to help understanding of our algorithm. For all few-shot classification experiments, we use the ProtoNet-like parameter-free NCC head in our NIW-Meta. Some important implementation details on the SGLD iterations for quadratic approximation of the local episodic optimisation include: we have either 3 steps without burn-in (for large-scale backbones ViT) or 5 steps with 2 burn-in steps (for smaller backbones ConvNet, ResNet-18, and CNP). Before starting SGLD iterations, the network is initialised with the current model parameters $m_0$. For reliable variance estimation of $\overline{A}_i$, a small regulariser is added to the diagonal entries of the variances.

For the standard benchmarks with ConvNet/ResNet backbones, we follow the standard protocols of (Wang et al., 2019; Mangla et al., 2020; Zhang et al., 2021): With 64/16/20 and 391/97/160 train/validation/test class splits for *mini*ImageNet and *tiered*ImageNet datasets, respectively, the images are resized to 84 pixels. We initialise the $m_0$ parameters from the pretrained models: checkpoints from (Wang et al., 2019) for Conv-4 and ResNet-18 and checkpoints from (Mangla et al., 2020) for WRN-28-10. With the stochastic gradient descent (SGD) optimizer, we set momentum 0.9, weight decay 0.0001, and initial learning rate 0.01 for *mini*ImageNet and 0.001 for *tiered*ImageNet. We have learning rate schedule by reducing the learning rate by the factor of 0.1 at epoch 70.

For the large-scale ViT backbones, we utilise the code base from (Hu et al., 2022). We use the self-supervised pretrained checkpoints from (Caron et al., 2021) to initialise the $m_0$ parameters. The CIFAR-FS dataset is formed by splitting the original CIFAR-100 into 64/16/20 train/validation/test classes. For training, we run 100 epochs, each epoch comprised of 2000 episodes. We follow the same warm-up plus cosine annealing learning rate scheduling as (Hu et al., 2022). For test evaluation, we have 600 episodes from the test splits.

For the few-shot regression experiments with ShapeNet datasets, we basically follow all experimental settings and CNP/ANP network architectures from (Gao et al., 2022). For instance, in the ShapeNet-1D dataset, we run our algorithm for $500K$ iterations with learning rate $10^{-4}$ where each batch iteration consists of 10 episodes. The CNP backbone, for instance, in the Distractor dataset case, has a ResNet image encoder and a linear target encoder, where the concatenated instance-wise embeddings then go through a three-layer fully connected network followed by max pooling. The decoder has a similar architecture and converts the support set embedding and a query image into a target label. For the conv-net plus ridge-regression head backbone (C+R) tested for our method, the conv-net feature extractors are formed by taking the encoder parts of the CNP architectures in (Gao et al., 2022) while discarding the pooling operations and decoders. Also the ridge-regression L2 regularisation coefficient is set to $\lambda = 1.0$ for all datasets.

## E  COMPUTATIONAL COMPLEXITY, RUNNING TIME AND MEMORY FOOTPRINT

Although we have introduced a principled Bayesian model/framework for FSL with solid theoretical support, the extra steps introduced in our training/test algorithms appear to be more complicated than simple feed-forward workflows (e.g., ProtoNet (Snell et al., 2017)). To this end, we have analysed the time complexity of the proposed algorithm contrasted with ProtoNet (Snell et al., 2017). For fair comparison, our approach adopts the same NCC head on top of the feature space as ProtoNet.

---

[4]https://github.com/facebookresearch/higher

Table 8: (Per-episode) Time complexity of our NIW-Meta vs. ProtoNet. We denote by $F_D$ and $B_D$ the forward-pass and backpropagation times with data $D = S$upport or $Q$uery. In our algorithm, $M_L$, $M_V$, and $M_S$ indicate the numbers of SGLD iterations, test-time variational inference steps for (13) or (63,64), and the number of test-time model samples $\theta^{(s)}$, respectively. The costs required for reparametrised sampling in model space and regulariser computation in (11) or (62) are denoted by $O(d)$ where $d$ = number of backbone parameters.

| | Training time | Test time |
|---|---|---|
| NIW-Meta (Ours) | $(F_S+F_Q+B_Q)\cdot(M_L+1)$ 
 $+ O(d)$ | $(F_S+B_S)\cdot M_V +$ 
 $(F_S+F_Q)\cdot M_S + O(d)$ |
| ProtoNet | $F_S+F_Q+B_Q$ | $F_S+F_Q$ |

Table 9: Effect of the finite number of episodes ($N$) on the generalization error gap in terms of the sample complexity gap $\log(2\sqrt{nN}/\delta)/(nN)$ in (21). Here $\delta = 0.001$.

| $N$ (#episodes) | $n$ (#shots $\times$ #ways) | Sample complexity error gap ($\downarrow$) |
|---|---|---|
| $10^3$ | $1\times5$ / $5\times5$ | $2.4\times10^{-3}$ / $5.0\times10^{-4}$ |
| $10^4$ | $1\times5$ / $5\times5$ | $3.0\times10^{-4}$ / $5.5\times10^{-5}$ |
| $10^5$ | $1\times5$ / $5\times5$ | $2.8\times10^{-5}$ / $6.0\times10^{-6}$ |

The computational complexity is summarised in Table 8. Despite seemingly increased complexity in the training/test algorithms, our method incurs only constant-factor overhead compared to the minimal-cost ProtoNet.

As we claimed in the main paper, one of the main drawbacks of MAML (Finn et al., 2017) is the computational overhead to keep track of a large computational graph for inner gradient descent steps. Unlike MAML, our NIW-Meta has a much more efficient episodic optimisation strategy, i.e., our local episodic optimisation only computes the (constant) first/second-order moment statistics of the episodic loss function without storing the full optimisation trace.

To verify this, we measure and compare the memory footprints and running times of MAML and NIW-Meta on two real-world classification/regression datasets: *mini*ImageNet 1-shot with the ResNet-18 backbone and ShapeNet-1D with the ConvNet backbone. The results in Fig. 5 show that NIW-Meta has far lower memory requirement than MAML (even smaller than 1-inner-step MAML) while MAML suffers from heavy use of memory space, nearly linearly increasing as the number of inner steps. The running times of our NIW-Meta are not prohibitively larger compared to MAML where the main computational bottleneck is the SGLD iterations for quadratic approximation of the local episodic optimisation. We tested two scenarios with the number of SGLD iterations 2 and 5, and we have nearly the same (or even better) training speed as the 1-inner-step MAML.

## F    TRAINING STABILITY AND IMPACT OF NUMBER OF TRAINING EPISODES

In our theoretical analysis of the generalisation error (Sec. 5 in the main paper and our proofs in Sec. A), we regard the number of training episodes $N$ as infinity. In practice, $N$ is finite, but large enough ($N \sim 100K$ in typical FSL), and we simply take it as **infinite** $N$ for mathematical convenience (e.g., to have the first KL term in (5) vanish; reduction to the task population mean from (29) to (30)). To see the effect of finite $N$ on the generalization performance, we exemplify several typical $N$ values and corresponding generalization (sample complexity) error gaps (21) in Table 9. We see that even for relatively small $N$, error gaps are small/negligible.

In addition, we investigate the impact of the number of training episodes $N$ on training stability. We illustrate it in Fig. 6, which shows that our method works stably well even for a small number of initial episodes, convergence being as fast as ProtoNet with far better generalization performance.

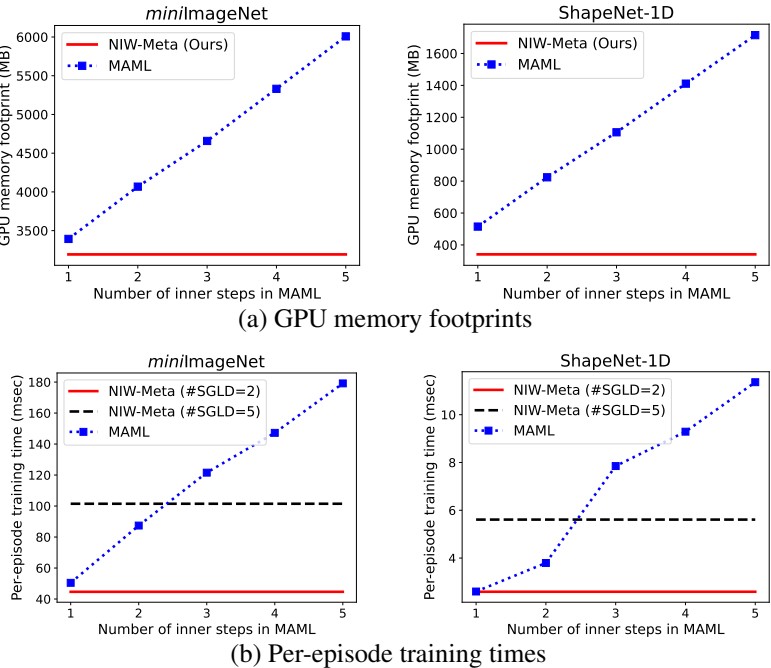

(a) GPU memory footprints

(b) Per-episode training times

Figure 5: Computational complexity of MAML (Finn et al., 2017) and our NIW-Meta. (a) GPU memory footprints (in MB) for a single batch. (b) Per-episode training times (in milliseconds). For our NIW-Meta models, the time for the number of burn-in steps (2 steps in this case) is also included. That is, `NIW-Meta(#SGLD=2)` runs $2 + 2$ SGLD iterations, and `NIW-Meta(#SGLD=5)` runs $5 + 2$, respectively, compared to MAML with $1 \sim 5$ inner iterations. We use the ResNet-18 backbone for *mini*ImageNet in 1-shot classification and the ConvNet backbone for ShapeNet-1D regression (10 episodes per batch).

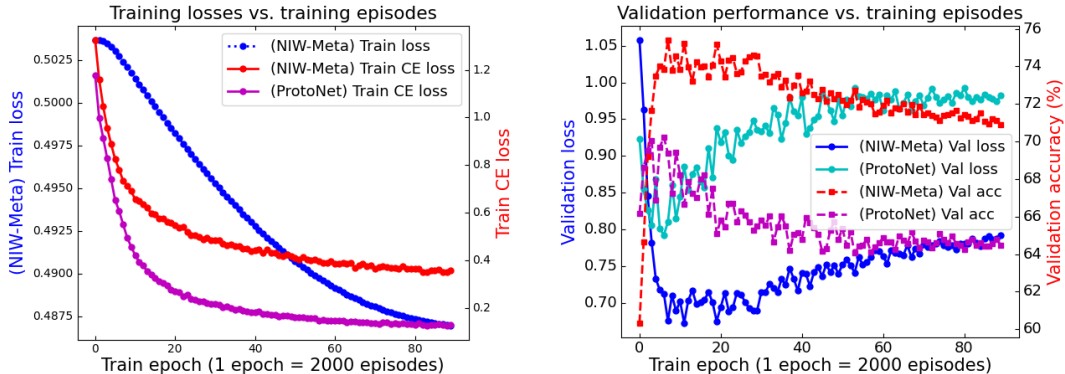

Figure 6: CIFAR 1-shot learning with ViT (DINO/s). (Left) Training losses vs. training episodes (2000 episodes per epoch). We plot the training loss of our NIW-Meta (i.e., (11)) in blue, and super-impose the training cross-entropy loss in red, where the latter is comparable to ProtoNet's training CE loss in magenta. We see that our NIW-Meta training is pretty stable, where the convergence speed is as fast as ProtoNet. (Right) Performance on the validation set as training episodes increases. The validation losses of our NIW-Meta (blue) and ProtoNet (cyan) are comparable, while we also compare the validation accuracy of our NIW-Meta (red) against ProtoNet (magenta).

Table 10: Predictive log-marginal-likelihood scores (LMLHD) on the Sine-Line dataset. The higher the better uncertainty quantification and generalisation.

| Context size | 1 | 2 | 4 | 6 | 8 | 12 | 16 |
|---|---|---|---|---|---|---|---|
| (Volpp et al., 2023) | -18.38 | -15.98 | -13.69 | -12.74 | -11.75 | -11.14 | -10.23 |
| NIW-Meta (Ours) | -17.02 | -14.74 | -12.60 | -8.14 | -3.51 | -1.55 | -1.33 |

# G  ADDITIONAL DISCUSSIONS

## G.1  COMPARISON TO BAYESIAN NEURAL PROCESSES

An interesting question is what are the relative merits of a complete Bayesian treatment but with restricted Gaussian forms as in our model, versus a shallow Bayesian but GMM-like rich variational forms such as (Volpp et al., 2023). We do not know which is absolutely better than the other. Further theoretical and empirical study needs to be carried out in this regard. But we have some experimental evidence as demonstrated in our paper, the comparison with MetaQDA (Zhang et al., 2021) shallow Bayesian approach that only places prior distribution on the model head parts while freezing the feature extractor (Table 2, 3, 5). As shown, our complete Bayesian treatment outperformed it, in both test generalisation performance and uncertainty calibration. This is one supporting evidence of why a complete Bayesian treatment could be more promising than a shallow Bayesian treatment.

For the experimental demonstration, we have done additional experiments on the Sine-Line dataset to report the predictive marginal test likelihood score, which is directly comparable to the shallow Bayesian embedding models, esp., the Bayesian Neural Process model of (Volpp et al., 2023). The predictive log-marginal-likelihood scores (LMLHD) are shown in Table 10. We try to match the experimental settings from (Volpp et al., 2023) so that the results are comparable. More specifically, we test on 256 tasks, each of which consists of 64 samples with a varying number of context samples (1, 2, 4, 6, 8, 12 and 16). The number of posterior samples used to compute/approximate the predictive marginal likelihood is 1024. As shown, our NIW-Meta has higher scores, especially for larger context size, implying that capturing uncertainty in full model parameters is important for generalisation capability.

## G.2  JUSTIFICATION OF MODEL AND ALGORITHM CHOICES

**Motivation for distributional estimate** $q(\phi)$**.**    If $\phi$ were directly linked to the observed data $D_i$'s in our graphical model Fig. 1, then $\phi$ can tend to be determined deterministically, as the number of data $N$ becomes large. However, $\phi$ is linked to *latent variables* $\theta_i$'s, so the belief on $\phi$ also needs to capture and accumulate the uncertainty in $\theta_i$'s, which amounts to marginalising out the $\theta_i$ variables. So, it may not not be appropriate to treat the posterior for $\phi$ as a delta function (0 uncertainty), and it is better to follow the Bayesian inference principle, i.e., let the posterior be computed from the observed evidence.

**Why distributional estimate** $q(\phi)$ **if we only use the mode of** $q(\phi)$ **at meta-test time.**    Using the mode of $q(\phi)$ is only for practical convenience and simplicity. Although we used the mode, the distributional form $q(\phi)$ would take into account uncertainty in its optimisation, and thus lead to a different solution from the deterministic one.

**Why not the same inference approach for meta testing as meta training.**    For meta testing, we may attempt to solve the optimisation problem similar to (7) or (8) for meta training with $L_0$ fixed, which also results in the same meta test solution as our derivation in Sec. 3.2. This can also be verified by inspecting the similarity between (13) and (7) or (8). However, this approach requires that $L_0$ be fixed at the trained value, which we do not know for sure in the pure optimisation perspective. In Sec. 3.2, we aimed to derive the meta-test optimisation problem from the Bayesian perspective from the outset, which offers us a reasonable justification for why $L_0$ can be fixed.

**Quality of the quadratic approximation.**    In (9), we have made the quadratic approximation for the negative likelihood, $-\log p(D_i|\theta) \approx \frac{1}{2}(\theta - \overline{m}_i)^\top \overline{A}_i(\theta - \overline{m}_i) + \text{const}$. To see the quality of this approximation, we empirically evaluated the true value (the left hand side of the approximation) and the quadratic approximation (the right hand side) on the *mini*ImageNet 5-shot dataset with

the ResNet-18 backbone. We take random 10 perturbations of $\theta$ around the mode/mean $\overline{m}_i$ with perturbation radius $0.1$. The relative error of the quadratic approximation is $0.0050 \pm 0.0004$. This shows that the negative log-likelihood is well approximated by our quadratic function in the vicinity of the mode/mean.

### G.3 LIMITATIONS AND FUTURE WORKS

**Limitations.** 1) Our NIW-Meta introduces some extra hyperparameters (e.g., the number of SGLD iterations, the number of burn-in steps). These are currently estimated empirically, but a more rigorous study on how to select them automatically needs to be addressed. 2) Although it is empirically verified that our quadratic episodic loss optimisation is effective, more theoretical analysis on the quality of this approximation as well as its impact on the final results, needs to be done.

**Future works.** We have quite an extensive evaluation on popular few-shot classification and regression benchmarks. However, we would like to evaluate our approach on new emerging applications of few-shot learning such as efficient learning of the implicit neural representations such as NeRF, e.g., (Tancik et al., 2021).

