# OpenReview forum: "A Hierarchical Bayesian Model for Few-Shot Meta Learning"
_ICLR.cc/2024/Conference — ICLR 2024 spotlight_

### Official Review · Reviewer_Whmq · 2023-10-27

**Soundness:** 3 good
**Presentation:** 3 good
**Contribution:** 3 good
**Rating:** 6
**Confidence:** 4

**Summary:**

This work presents a hierarchical Bayesian model aimed at addressing the few-shot meta-learning problem. The task-specific parameters are influenced by a higher-level global random variable, responsible for capturing shared information across tasks. Within the framework, the prediction tasks on new tasks is viewed as a Bayesian inference problem.

**Strengths:**

The proposed method leverages conjugate properties to provide closed-form solutions for both task-specific and task-agnostic updates. Additionally, this method seamlessly integrates with a wide range of existing neural few-shot learning meta-learners, enhancing its versatility within the field. The empirical results are robust, showcasing improved accuracy and calibration performance across various benchmarks, encompassing tasks related to classification and regression.

**Weaknesses:**

I have partially reviewed the article, particularly focusing on the derivation of mathematical formulas, and the technical aspects appear to be solid. I did not identify any significant errors or weaknesses. The only minor concern is that the overall approach and the core ideas behind this work are somewhat similar to most Bayesian meta-learning approaches in the context of few-shot learning. These methods generally utilize priors to relate the parameters of multiple tasks, and they apply variational inference for posterior inference on task-level or meta-level parameters. In this regard, the level of innovation in the article may be somewhat limited.

**Questions:**

N/A

---

> ### Author Response · Authors · 2023-11-20
> **Responses to Reviewer Whmq**
>
> > **1. Similarity to most Bayesian meta-learning approaches in the context of few-shot learning.**
>
> The novelty of our work compared to other Bayesian approaches is four folds:
>
> * 1) We capture uncertainty of *entire* model, including the full feature extractor, by treating every weight as a random variable and placing priors on them. Most existing Bayesian meta-learners only capture uncertainty of a subset of weights (EG: VERSA, MetaQDA, DKT) and rely on deterministic feature extractors.
>
> * 2) Our *Hierarchical* Bayesian framework distinctly captures uncertainty at both the task ($\theta$) and dataset ($\phi$) levels. Some existing Bayesian attempts are not hierarchical and do not separately model these sources of uncertainty. This significance of hierarchical distinction is explored intuitively in Sec 4.
>
> * 3) We achieve hierarchical Bayesian modeling of the full neural architecture (1+2 above) with low computational overhead thanks to the proposed closed-form episodic loss quadratic approximation. Our Meta-NIW is more efficient than both (non-Bayesian) MAML, and  prior hierarchical Bayesian attempts such as PMAML. PMAML has only been demonstrated on small models and datasets, while we achieve full Bayesian uncertainty modeling on large VITs and challenging benchmarks such as ShapeNet2D and tieredImageNet.
>
> * 4) We provide rigorous theoretical analysis to support the generalisation capability of the provided algorithm (Sec 5). This guarantee is also a novel result to our knowledge.

---

> > ### Comment · Reviewer_Whmq · 2023-11-23
> > **Reply to rebuttal**
> >
> > Thanks for the authors' rebuttal. It addressed most of my concerns, I will keep my original score and incline to accept it.

---

### Official Review · Reviewer_EN3o · 2023-10-29

**Soundness:** 2 fair
**Presentation:** 3 good
**Contribution:** 2 fair
**Rating:** 6
**Confidence:** 4

**Summary:**

The paper studies the problem of few-shot learning in a model-based meta-learning setting. The authors employ a hierarchical Bayesian meta-model, variants of which have been studied extensively in the literature [1-10]. They parametrize the model by Gaussian priors for the task-specific variables and conjugate Gaussian-inverse-Wishart hyper-priors for the task-global variables, and consider various neural network based likelihood models from prior work. The authors propose a novel variational inference procedure to meta-learn the task-global posterior, which notably does not employ task-amortization in contrast to most prior work. Similarly, they determine the meta-test task-posterior using a variational approach starting from a mode of the task-global posterior. The authors include a toy experiment to demonstrate the relevance of hierarchical modeling for few-shot meta-learning. Furthermore, they evaluate their approach against several competing methods on standard benchmarks like meta-image classification, meta-regression, and pose estimation.


-------------------------


I increased my score from 5 to 6 after reading the author's response.

**Strengths:**

The paper is mostly well-written and presents a conceptually interesting approach for inference in hierarchical Bayesian meta-models. The derivation of the inference procedures for the task-global variables at meta-train time as well as the task-specific variables at meta-test time appears to be executed well, but gets technically quite dense at times and leaves open several important points, cf. weaknesses. While the toy experiment in Sec. 4 seems to be unrelated to the novel inference procedure developed in Sec. 3, it can serve as a nice pedagogical illustration of the efficiency of the known hierarchical Bayesian modeling approach for few-shot learning.

**Weaknesses:**

My main concern is that central claims of the paper are not sufficiently supported by theoretical arguments nor by experimental evidence, so I encourage the authors to provide further justification:
- The authors claim in the abstract to "propose a novel hierarchical Bayesian model for the few-shot meta learning problem". This is not accurate as the model and variants thereof have been studied extensively in the literature [1-10].
- The authors claim to provide "the first complete hierarchical Bayesian treatment of the few-shot deep learning problem". I ask the authors to elaborate on this claim.
  - In fact, as noted above, the model has been considered before for few-shot meta learning by a range of papers, all of which propose different parametrizations and inference approaches. The related work section mentions a few of them, but describes them as improper solutions of the Bayesian few-shot learning problem.
  - In particular, the authors claim that a Bayesian treatment of "only a small fraction" of the model's weights "considerably limits the benefits from uncertainty modeling". The authors cite (among others) and compare to the conditional neural process [11], which is a conditional model and, thus, not a Bayesian treatment of the few-shot learning problem, as well as to the attentive neural process [12], which has been shown in [6] to  revert to a mainly deterministic approach due to the deterministic, attentive computation path. Therefore, I do not consider these methods suitable baselines to support the author's claim, i.e., to show that their approach yields improved uncertainty modeling.
  - Unfortunately, the authors neither discuss nor compare to the Bayesian neural process versions proposed in [3] and further studied in, e.g., [4-6]. I consider such types of neural process based approaches the main competitors to the proposed inference scheme. Therefore, I ask the authors to discuss in which sense their approach is more "complete", and also to provide experimental evidence for their claim. In fact, performing inference over "only a small fraction" of the model's parameters opens the door for sophisticated variational inference methods and expressive variational approximations, which I consider a major benefit of such approaches [4,6]. In contrast, "complete hierarchical treatments" are typically restricted to simple Gaussian variational distributions (as also used in the proposed approach) due to the high dimensionality of the parameter space. I ask the authors to theoretically motivate and discuss, as well as to experimentally demonstrate that performing inference over all model parameters still outweighs these downsides.

Furthermore, as said above, I have doubts regarding the empirical evaluation. In particular, the authors claim that their approach is superior in terms of uncertainty modeling, cf. above, but I do not consider their experimental approach suitable to support this claim:
- Most of the experiments are classification tasks on which the authors report accuracy. I'd ask the authors to elaborate in which sense these experiments and metrics really measure uncertainty quantification.
- I acknowledge that the authors measure calibration in Tab. 6, but still think that most of the presented experimental evidence is unsuitable to support the central claims of the paper.
- Therefore, I encourage the authors to provide more experimental evidence for their claims. I am have no strong opinion on what experiments shall be presented, as long as they are suitable for measuring the quality of predictive uncertainty quantification, e.g.,
  - provide results also in terms of predictive log marginal likelihood [2,4-6,10],
  - provide calibration measurements for further experiments, possibly also in terms of reliability diagrams [8],
  - provide further experiments that better measure uncertainty quantification, e.g.,
    - further probabilistic regression experiments (such as the presented line-sine),
    - experiments that require well-calibrated uncertainties for exploration-exploitation trade-off, e.g., Bayesian optimization or Bandits as in [3,6,8].

Lastly, I ask the authors to elaborate on some more technical issues as well as on some algorithmic choices of their proposed inference scheme that did not become fully clear to me:
- I'm a bit confused about the inference procedure. In Sec. 2 the authors describe the standard meta-learning setting which considers a "large number of episodes". By construction, the task-global parameters $\phi$ are informed by all this data, which is why related work often argues that $\phi$ is determined with good accuracy, which allows empirical Bayes estimates, e.g., [2-6]. I'd ask the authors to elaborate on the following:
  - What is the theoretical motivation for distributional estimate $q(\phi)$ of the posterior of $\phi$?
  - Furthermore, why do we need a distributional estimate $q(\phi)$ if at meta-test time we only use the mode of $q(\phi)$?
  - Along the same lines, I wonder why the authors do not use the same inference approach for the $L_*$ at meta-test time (for a test episode indexed by \*) as they use during meta training, i.e., why don't they use the (now fixed) $L_0$ obtained from meta-training and solve Eq. (8) for $m_*$, $V_*$?
- I'd ask the authors to add some more details on whether the two-stage optimization approach of (i) reducing the optimization problem over $(L_{1:N}, L_0)$ to one over $L_0$ alone by solving for $L_i^*$ for fixed $L_0$ and (ii) solving for $L_0$ is guaranteed to find the same solution as the original joint optimization problem.
- I'd be interested in some more architectural details about how the combination of the authors proposed approach with set-based backbones like CNP works, cf. end of Sec. 3.

Further assorted issues:
- Not all results come with error bounds. How were the error bounds of the presented results computed?
- The boldfaced results do not always seem to be better within error bounds. What is the author's "protocol" to boldface results?
- Table 3/4 captions do not say that classification accuracy is reported.

To summarize: I consider the proposed inference approach an interesting and technically sound contribution that is relevant for the community. Nevertheless, I recommend to reject the paper in its current form, because I find that central claims the authors make require further discussion and experimental evidence. I'm looking forward to the author's response and I'm willing to increase my score after the rebuttal period in case my concerns are adressed.

[1] Bakker & Heskes, "Task Clustering and Gating for Bayesian Multitask Learning", JMLR 2003

[2] Gordon et al., "Meta-Learning Probabilistic Inference for Prediction", ICLR 2019

[3] Garnelo et al., "Neural Processes", ICML Workshop 2018

[4] Volpp et al., "Bayesian Context Aggregation for Neural Processes", ICLR 2021

[5] Wang & van Hoof, "Double Stochastic Variational Inference for Neural Processes with Hierarchical

Latent Variables", ICML 2020

[6] Volpp et al., "Accurate Bayesian Meta-Learning by Accurate Task Posterior Inference", ICLR 2023

[7] Amit & Meir, "Meta-Learning by Adjusting Priors Based on Extended PAC-Bayes Theory", ICML 2018

[8] Ravi & Beatson, "Amortized Bayesian Meta-Learning", ICLR 2019

[9] Grant et al., "Recasting Gradient-Based Meta-Learning as Hierarchical Bayes", ICLR 2018

[10] Finn et al., "Probabilistic Model-Agnostic Meta-Learning", NeurIPS 2018

[11] Garnelo et al., "Conditional Neural Processes", ICML 2018

[12] Kim et al., "Attentive Neural Processes", ICLR 2019

**Questions:**

cf. weaknesses

---

> ### Author Response · Authors · 2023-11-20
> **Responses to Reviewer EN3o (Part 1)**
>
> > **1. The claim of *”A novel hierarchical Bayesian model for FSL”* in the abstract.**
>
> Thanks for the feedback. We agree that several other papers took hierarchical Bayesian perspectives on FSL. Our contribution lies in a specific parametrization and inference algorithm that allows fast and effective learning of such HBMs in large neural networks. We have adjusted our wording accordingly in the revised paper.
>
>
> > **2. Describing some of the existing methods as improper solutions of the Bayesian few-shot learning problem.**
>
> Thanks for the feedback. We felt there were some limitations in existing attempts, such as being Bayesian but non-hierarchical, or only modeling a subset of the weights in a hierarchical Bayesian manner. However, we will carefully revise and tone down our discussion to avoid overclaiming. And of course, we have cited the mentioned papers that we did not include initially. Please check our revised paper (e.g., end of Sec. 6 Related Work).
>
>
> > **3. The claim of "*Bayesian treatment of only a small fraction of the network limits the benefits from uncertainty modeling*". About CNP/ANP baselines to support claims on improved uncertainty modeling.**
>
> With regard to improved uncertainty modeling, we mention the CNP/ANP for their non-Bayesian modeling. In our evaluation on uncertainty calibration (Table 5), we did not compare ours to CNP/ANP but mainly to MetaQDA, a Bayesian model that is state of the art among those that only provide a Bayesian treatment of the output layer (VERSA, Gordon ICLR’19 and DKT Patacchiola’20, etc). The comparison here supports the claim that our Bayesian treatment of the entire network is beneficial.
>
>
> > **4. Comparison to Bayesian Neural Process models. Relative merits of a complete Bayesian treatment.**
>
> Regarding the question on the relative merits of a complete Bayesian treatment but with restricted Gaussian forms versus a shallow Bayesian but GMM-like rich variational forms – We do not know which is absolutely better than the other. Further theoretical and empirical study needs to be carried out in this regard. But at this point, we have some experimental evidence, the  comparison with MetaQDA shallow Bayesian approach that only places prior distribution on the model head parts while freezing the feature extractor (Table 2,3,5). As shown, our complete Bayesian treatment outperformed it, in both test generalisation performance and uncertainty calibration. This is one supporting evidence of why a complete Bayesian treatment could be more promising than a shallow Bayesian treatment.
>
> For the experimental demonstration, we have now done additional experiments on the Sine-Line dataset to report the predictive marginal test likelihood score, which is directly  comparable to the shallow Bayesian embedding models (e.g., [6]'s Bayesian NP model). Please see our table in response to Q7 below (or the second table in our responses to all reviewers).
>
>
> > **5. Most of the experiments are classification tasks. Why ECE metric?**
>
> Our main target problem is few-shot meta learning (FSL). And those classification experiments (miniImageNet, tieredImageNet) are recognised as the mainstream benchmarks for FSL. We follow prior Bayesian FSL methods (eg: MetaQDA - Zhang ICCV’21; DKT - Patacchiola NeurIPS’20; Amortised Meta - Ravi ICLR’19) to assess uncertainty quantification via ECE and R-ECE on these standard benchmarks.
>
>
> > **6. Calibration measures in Table 6.**
>
> Please refer to (Tran et al. 2020) and (Cui et al. 2020) . Together with the test likelihood score, we believe that the ECE score is the mainstream measure for uncertainty calibration (Hospedales et al., 2022; Wang et al., 2020a). We go beyond the evaluations of existing learners that mainly focused on uncertainty quantification in classification (ECE, Tab 5), and also report uncertainty quantification in regression (Tab 6).
>
> > **7. Provide results in terms of predictive log marginal likelihood.**
>
> As requested by the reviewer, we have done experiments with the SineLine dataset to report the predictive log-marginal-likelihood scores (LMLHD). We try to match the experimental settings from [6] so that the results are comparable. More specifically, we test on 256 tasks, each of which consists of 64 samples with a varying number of context samples (1,2,4,6,8,12, and 16). The number of posterior samples used to compute/approximate the predictive marginal likelihood is 1024. The results are as follows (the higher the better uncertainty quantification and generalisation). As shown, our NIW-Meta has higher scores, especially for larger context size, implying that capturing uncertainty in full model parameters is important for generalisation capability.
>
> | Context size | 1 | 2 | 4 | 6 | 8 | 12 | 16 |
> | ------------- |-----|-----|-----|-----|-----|-----|-----|
> | (Volpp et al., ICLR’23) | -18.38 | -15.98 | -13.69 | -12.74 | -11.75 | -11.14 | -10.23 |
> | NIW-Meta (Ours) | -17.02 | -14.74 | -12.60 | -8.14 | -3.51 | -1.55 | -1.33 |

---

> ### Author Response · Authors · 2023-11-20
> **Responses to Reviewer EN3o (Part 2)**
>
> > **8. What is the theoretical motivation for distributional estimate $q(\phi)$?**
>
> If $\phi$ were directly linked to the observed data $D_i$’s, then it is true that $\phi$ can tend to be determined deterministically, as the number of data $N$ becomes large.  However, $\phi$ is linked to *latent variables* $\theta_i$’s, so the belief on $\phi$ also needs to capture/accumulate the uncertainty in $\theta_i$’s, which amounts to marginalising out the $\theta_i$ variables. So, it may not not be appropriate to treat the posterior for $\phi$ as a delta function (0 uncertainty), and it is better to follow the Bayesian inference principle, i.e., let the posterior be computed from the observed evidence.
>
>
> > **9. Why do we need a distributional estimate $q(\phi)$ if at meta-test time we only use the mode of $q(\phi)$?**
>
> Using the mode of $q(\phi)$ is only for practical convenience and simplicity. Although we used the mode, the distributional form $q(\phi)$ would take into account uncertainty in its optimisation, and thus lead to a different solution from the deterministic one.
>
>
> > **10. For meta testing, why the authors do not use the same inference approach as they use during meta training?**
>
> The method that the reviewer said also results in the same meta test solution, and this can also be verified by inspecting the similarity  between Eq.(13) and Eq.(7 or 8). However, the mentioned approach requires that $L_0$ be fixed at the trained value, which we do not know for sure in the pure optimisation perspective. In Sec. 3.2, we aimed to derive the meta-test optimisation problem from the Bayesian perspective from the outset, which offers us a reasonable justification for why $L_0$ can be fixed.
>
>
> > **11. Is the two-stage optimization approach (described in the paragraph before Eq.(7)) guaranteed to find the same solution as the original joint optimization problem?**
>
> Yes, the two-stage approach guarantees to find the same solution as the original joint problem.  As we mentioned in the paper, this is essentially the same as $\min_{x,y} f(x,y) = \min_x f(x,y^*(x))$ where $y^*(x) = \arg\min_y f(x,y)$ for each given x – by the very definition of the joint minimisation problem.
>
> > **12. Some more architectural details about how the combination of the authors proposed approach with set-based backbones like CNP works, cf. end of Sec. 3.**
>
> We basically follow the architectures from (Gao et al. 2022) https://arxiv.org/pdf/2203.04905.pdf
> For instance, for the ShapeNet1D case, the networks to which our NIW-Meta was applied are:
>
> * CNP:
>     * Rsupp = enc2( enc1(Xsupp), Ysupp )    where enc1 = 4-layer conv net, enc2 = 2-layer MLP
>     * Zsupp = max-pool(Rsupp)
>     * Yqry_prediction = dec( enc1(Xqry), Zsupp )    where dec = 3-layer MLP
>
> * ANP:
>     * We replace “max-pool” in CNP by “multi-head-attn( enc1(Xsupp), Rsupp, enc1(Xqry) )”
>
> * C+R (Conv-Net + Ridge regression head):
>     * Yqry_prediction = W* @ \phi(Xqry) where \phi() is 4-layer conv + 2-layer MLP, and
>     * W* = argmin_W || W @ \phi(Xsupp) - Ysupp ||^2 + \lambda*||W||^2 (a  closed-form solution)
>
> For further details and other datasets, please refer to (Gao et al. 2022).
>
>
> > **13. Not all results come with error bounds. The boldfaced results. Table 3/4 captions do not say that classification accuracy is reported.**
>
> * Error bars:  Now we have added the SEMs for the results in Table 3 to show the statistical significance. Please see the revised paper for the full results and also the table below for only DINO/s CIFAR. Even though for some tasks there are some overlaps of the error intervals between our proposed approach and the competing methods, overall our approach is superior to the existing methods with statistical significance to some extent.
>
> | CIFAR, DINO/s | 1-shot | 5-shot |
> | ------------- |:-------------:| -----:|
> | ProtoNet      | $81.1 \pm 0.29$ | $92.5 \pm 0.13$ |
> | MetaOpt      | $70.2 \pm 0.22$ | $84.1 \pm 0.27$ |
> | MetaQDA    | $77.2 \pm 0.34$ | $90.1 \pm 0.18$ |
> | **NIW-Meta (Ours)** |  $\bf{82.8 \pm 0.26}$ | $\bf{92.9 \pm 0.11}$ |
>
>  * Boldfaced results: We marked the ones with the largest average values.
>
> * For Table 3/4: In our revised paper, we have mentioned that they are classification accuracy. Thank you for pointing this out.

---

> > ### Comment · Reviewer_EN3o · 2023-11-22
> > **Thanks for the answer**
> >
> > I thank the authors for their detailed response! I think the clarifications further improve the paper, so I increase my score. I encourage the authors to also include their answers (4),(8),(9),(10) in the revised version of their manuscript, as I'm convinced that this discussion is valuable for the community.

---

> > > ### Author Response · Authors · 2023-11-23
> > > **Thank you very much!**
> > >
> > > Thank you very much for the suggestions! We have added the answers (4),(8),(9),(10) in the revised manuscript.

---

### Official Review · Reviewer_kRtC · 2023-11-08

**Soundness:** 4 excellent
**Presentation:** 4 excellent
**Contribution:** 4 excellent
**Rating:** 8
**Confidence:** 4

**Summary:**

This paper proposes a novel hierarchical Bayesian model for the few-shot meta-learning problem. They consider episode-wise random variables to model the generation process, where these local random variables are governed by higher-level global random variables. The global variable captures information shared across episodes while controlling how much the model needs to be adapted to new episodes in a principled Bayesian manner. Prediction on a novel task can be seen as a Bayesian inference problem. They propose a Normal-Inverse-Wishart model, for which establishing the local-global relationship becomes feasible due to the approximate closed-form solutions for the local posterior distributions.

**Strengths:**

1. This paper actually proposes a unified framework for hierarchical Bayesian learning for few-shot problems.
2. They interpret the model in a way that MAML, ProtoNet and Reptile are special cases of the hierarchical Bayesian learning.
3. A detailed introduction and explanation of the methods can be seen in the paper,  also good experimental results have been obtained.

**Weaknesses:**

1. Please provide more background knowledge for the Normal-Inverse-Wishart model, and why this is justified in an application for this model.
2. This model is similar to hierarchical recurrent VAE, if so, what are the advantages of this model?
3. More general interpretation is needed (in the sense that not only in the Bayesian learning but for deep learning in general) so that the readers can benefit more.

**Questions:**

1. From Figure 3, (c) seems can be modelled via recurrent VAEs. which were proposed before, so what are the novelties of this paper on the few-shot learning task?
2. If possible, not only few-shot learning but other related tasks could be tested to validate the generalisation ability of the model.

---

> ### Author Response · Authors · 2023-11-20
> **Responses to Reviewer kRtC**
>
> > **1. More background knowledge for the NIW model and its justification.**
>
> The main reason why NIW model is adopted is that it allows a closed-form solution to the approximate episodic optimisation problem due to conjugacy with Gaussian. This is what enables us to achieve fast task-wise adaptation in (episodic) few-shot meta learning. This improved efficiency is part of what allows us to fully apply Bayesian learning to state of the art feature extractors to achieve excellent results, where many existing Bayesian attempts are restricted to relatively toy problems (PMAML, etc), or only a one Bayesian layer within a NNet (VERSA, MetaQDA, etc).
>
> Regarding the background knowledge, we recommend and refer to K. Murphy's MLPP book (https://probml.github.io/pml-book/book1.html), especially the probability and Bayesian statistics sections (eg, Ch.2, Ch.4.6, and Ch.A.8.3). It is overwhelming for us to make the paper fully self-contained by including all required background materials in it. Instead, we have cited these background materials in the revised paper.
>
>
> > **2. Advantage over hierarchical recurrent VAEs?**
>
> At first glance models following the Hierarchical Bayesian structures may look similar to each other. But the main difference is specific modeling and approximate inference strategies employed, in particular tailored for the specific FSL problem in this paper. We need fast task-wise adaptation, ie, fast solution for the inner variable variational optimisation, and to this end we came up with novel quadratic likelihood approximation under the NIW conjugate prior-posterior modeling.
>
> Another novelty is the theoretical analysis, which to the best of our knowledge has not been done for other HBMs.
>
>
> > **3.  Interpretation in the sense of deep learning in general.**
>
> One can find interesting interpretations from our discussions in Sec.3.1, where we interpret various non-Bayesian methods including MAML, ProtoNet, and Reptile as special cases of ours. That is, these non-Bayesian methods can be instantiated by dropping either stochasticity (uncertainty modeling) or prior-related terms in our Bayesian model.
>
>
> > **4. Generalisation ability of the model to tasks other than FSL?**
>
> Our approach can, in principle, be applicable to any other problems that involve multiple different data distributions (e.g., client/local data distributions in Federated or distributed learning). But we focus in this paper on the FSL problem mainly, leaving the application to other problems as future work.

---

### Official Review · Reviewer_PGLR · 2023-11-08

**Soundness:** 3 good
**Presentation:** 2 fair
**Contribution:** 3 good
**Rating:** 6
**Confidence:** 2

**Summary:**

The paper proposes a Bayesian framework to explicitly leverage the hierarchical structure of the generative process in few-shot learning. The Normal-Inverse-Wishart model-based variational inference and quadratic approximation allow efficient learning. The theoretical results link the essential values in the approximations to the ultimate performance.

**Strengths:**

1. Theoretical results on the PAC-Bayes-$\lambda$ bound and regression analysis reveal the convergence of the proposed method, though there is no convergence rate.

2. The proposed framework offers a unified interpretation for different established FSL methods.

3. The toy example in section 4 clearly motivates the proposed hierarchical model.

**Weaknesses:**

1. There are multiple approximations, most importantly the ELBO and the quadratic approximation, in the proposed framework. Though well-motivated, the lack of empirical evolution of the effectiveness of the approximations other than the downstream performance undermines the significance of the work. Especially given that the theoretical analysis in section 5 also relies on the quality of the approximation.

2. In section 7, the empirical results for some tasks only show average values rather than showing confidence intervals. And some superior performances lack statistical significance.

**Questions:**

Could the author comment on the choice of the parameters introduced by the proposed method, including the number of SGLD iterations, test-time variational inference steps for (13) or (63,64), and the number of test-time model samples?

---

> ### Author Response · Authors · 2023-11-20
> **Responses to Reviewer PGLR**
>
> > **1. Effectiveness of the multiple approximations, the ELBO and the quadratic approximation, in the proposed framework.**
>
> Both ELBO and the quadratic approximation are essential parts of our approach. ELBO is one of the few approximate methods to make Bayesian inference feasible, and is also used by other Bayesian methods such as VERSA (Gordon ICLR’19). And our quadratic approximation is what distinguishes it from other (hierarchical) Bayesian methods such as Probabilistic MAML (PMAML).
>
>
> > **2. Empirical results for some tasks only show average values without confidence intervals.**
>
> We now have added the SEMs for the results in Table 3 to show the statistical significance. Please see the revised paper for the full results and also the table in our official comments for all reviewers for only DINO/s CIFAR. Even though for some tasks there are some overlaps of the error intervals between our proposed approach and the competing methods, overall our approach is superior to the existing methods with statistical significance to some extent.
>
>
> > **3. Choice of the hyperparameters introduced by the proposed method.**
>
> Most hyperparameters are discussed and reported in Sec. D (Impl. details and experimental settings) of Appendix. For the test-time number of VI steps, we use 3 (mini/tiered-ImageNet), 2 (SineLine), and 20 (ShapeNets) iterations. The test-time number of posterior model samples is: 5 (mini/tiered-ImageNet and SineLine) and 2 (ShapeNets) samples.

---

### Official Review · Reviewer_ZG8Z · 2023-11-09

**Soundness:** 3 good
**Presentation:** 3 good
**Contribution:** 3 good
**Rating:** 6
**Confidence:** 3

**Summary:**

The manuscript proposes a hierarchical Bayesian meta learning model for the few-shot learning problem, where the Normal-Inverse-Wishart (NIW) model is used to model the globally shared variables and the individual episode-wise task-specific variables. Using the NIW-Gaussian conjugacy and the stochastic gradient Langevin dynamics (SGLD), the authors propose the approximate closed-form solutions for the local posterior distributions, which makes the proposed methods scalable to large backbone networks.

**Strengths:**

The manuscript proposes a complete hierarchical Bayesian model for the few-shot meta-learning problem and a scalable training algorithm.
The manuscript provides the theoretical analysis for the proposed method.

**Weaknesses:**

The authors claim that the proposed method shows the improved prediction accuracy and calibration performance.
- What aspect of the proposed method improves the prediction performance is not yet clearly stated in the text. The Bayesian treatment? It should be made more explicit and clearer in the text.
- It seems that the experiments do not include other Bayesian meta-learning methods. The hierarchical Bayesian model is the standard way to extend existing methods in Bayesian learning. I am not an expert on meta-learning but found several publications on this extension using simple keyword search. How about this paper [Reference]? I think that not comparing the proposed method with other Bayesian meta-learning methods limits the significance of the proposed method.
- The calibration performance is validated using only a single data set.

[Reference] Zhang, Z.; Li, X.; Wang, S. “Amortized Bayesian Meta-Learning with Accelerated Gradient Descent Steps” Appl. Sci. 2023, 13, 8653. https://doi.org/10.3390/app13158653

**Questions:**

1.	Regarding Section 5, what kind of information about the proposed method can we get from the final forms of the theorems, eq. (14) and eq. (15)?  For example, can we say anything about the choice of the hyperparameter \Lambda or about the relationship between the quality of the approximation of the posterior distributions and the generalization error bound?

2.	Regarding Figure 5, first the figures can be misleading because the computational complexities of your method look constant with the common parameter with MAML (which is not, the parameters for only MAML) in the current presentation. NIW-Meta, more precisely SGLD, requires multiple iterations as it skips the first burn-in iterations. How about comparing the actual training times of both methods when they are optimized to show the similar performance?

---

> ### Author Response · Authors · 2023-11-20
> **Responses to Reviewer ZG8Z**
>
> > **1. What aspect of the proposed method improves the prediction performance? The Bayesian treatment?**
>
> Yes. We attribute our good performance to the Bayesian nature of our method. More specifically due to the three factors:
>
> * **Uncertainty capturing**. Our ability to capture uncertainty is beneficial compared to non-Bayesian methods, but is shared with other Bayesian methods;
>
> * **Full Bayesian treatment**. We treat all parameters in the whole network as random variables in Bayesian inference (IE: capturing uncertainty everywhere instead of only at certain layers). This improves performance compared to those Bayesian methods that are only Bayesian about a subset of layers. EG: MetaQDA (Zhang, ICCV’21); VERSA (Gordon, ICLR’19); DKT (Patacchiola, NeurIPS’20); etc. Importantly, the minority of other methods that attempt a Bayesian treatment of every parameter incur a substantial computational overhead (eg: PMAML, Grant ICLR’18), which limits their applicability to  powerful modern neural networks required to achieve good performance. NIW-Meta's efficient Bayesian treatment enables us to benefit from both full uncertainty capturing together with state of the art feature extractors.
>
> * **Hierarchical Bayesian uncertainty capturing**. Our hierarchical Bayesian model captures distinct uncertainty effects within tasks/episodes, and across tasks/episodes. This distinction is illustrated in the toy experiment of Table 1 and Figure 2, 3. Several existing Bayesian methods (eg: VERSA, etc) are not hierarchical in this sense, and so do not benefit from this.
>
>
> > **2. Missing experimental comparison with other Bayesian meta-learning methods.**
>
> We did already compare with two Bayesian meta-learning methods: MetaQDA and PMAML. MetaQDA takes a limited Bayesian treatment (only one Bayesian layer in the NN), prioritizing the ability to scale to state of the art datasets and neural feature extractors and achieve state of the art results. MetaQDA already compares to, and beats, a range of prior Bayesian methods in their paper (VERSA - ICLR’19, DKT - NeurIPS20, etc), so we indirectly compare against even more Bayesian methods. Thus, we focus on comparing to MetaQDA as one of the strongest Bayesian SotA, which we outperform reliably (Tab 2, 3, 5).
> PMAML is a more complete Bayesian treatment, but does not scale to the large neural architectures required to achieve credible performance in vision tasks. Therefore we restrict our comparison to PMAML to few-shot regression problems (Sec 7.2).
>
> The FSL paper identified by the reviewer, (Zhang et al. 2023), is an extension of the amortized Bayesian inference (Ravi and Beatson, ICLR’19), which aims to accelerate the slow computation time of the original method. Comparing with these other Bayesian methods, our accuracy is significantly higher, and calibration is similar or better. Please see the table below.
>
> On miniImageNet (5-way, 1-shot) with Conv-4:
> |                 | Test accuracy |  ECE | MCE |
> | ------------- |:-------------:|:-----:|:-----:|
> | MAML | $47.0 \pm 0.59$ | 4.71 | 11.04 |
> | PMAML | $47.8 \pm 0.61$ | 4.72 | 8.56 |
> | (Ravi & Beatson, ICLR’19) | $45.0 \pm 0.60$ | **1.24** | 2.57 |
> | NIW-Meta (Ours) | ${\bf 56.8 \pm 0.59}$ | 1.47 | **2.04** |
>
>
> > **3. The calibration performance is validated using only a single data set.**
>
> Actually, we evaluated calibration on two datasets/tasks: Both miniImageNet (Tab 5) and Sine-Line (Tab 6). We have now also done new experiments to evaluate calibration in terms of the predictive marginal likelihood at test time. Please see our responses to **EN3o**’s Q.3(c).
>
>
> > **4. What kind of information about the proposed method can we get from the final forms of the theorems, Eq.(14) and Eq.(15)?**
>
> The theorems say that our hierarchical Bayesian modeling and our approximate posterior family choice (ie, NIW), are sufficiently good enough for provable test performance. That is, our ultimate optimisation objective Eq.(6), once optimised properly, can lead to a model that generalises well to unseen test episodes with the provided certificates.
>
>
> > **5. Comparing the actual training times in Fig. 5.**
>
> Figure 5 is actually already a fair comparison. Figure 5(b) already shows the full actual training times for competing methods. (Due to the far inferior performance of MAML, we are not able to compare the methods at similar performance.) However, we realise that there are some unclear points which may cause misunderstanding. For instance, in [Fig.5 (b) Left], we have two NIW-Meta models with different numbers of SGLD iterations (#SGLD=2 and #SGLD=5). The plots *already include* the time for both the number of burn-in steps, which is 2 burn-in steps in this case, and SGLD iterations (so, more precisely, they run 2+2 and 5+2 SGLD iterations, respectively, and compared to MAML with 1~5 inner iterations). It is our mistake not to mention this clearly enough, and we have clarified this in our revised paper.

---

### Official Review · Reviewer_wcX1 · 2023-11-10

**Soundness:** 3 good
**Presentation:** 3 good
**Contribution:** 4 excellent
**Rating:** 8
**Confidence:** 3

**Summary:**

This paper introduces a novel approach to few-shot meta-learning using hierarchical Bayesian models. In particular, a global random variable governs the shared information across different tasks, whereas local random variables model the generative process for a particular task. Using Normal-Inverse-Wishart distributions allows for closed-form expressions of the ELBO, allowing local episodic optimization. The authors show that many seminal few-shot learning approaches can be seen as special cases of the hierarchical Bayesian framework while their proposed algorithm does not require a full computation graph. The hierarchical Bayesian model is motivated by a toy example that highlights the need for both shared parameters across tasks and task-specific parameters. In addition, the authors provide generalization error bounds using the PAC-Bayesian framework and evaluate the proposed hierarchical framework on a range of regression and classification tasks.

**Strengths:**

- The paper presents the first comprehensive treatment of the hierarchical Bayesian framework. The framework is motivated using a simple toy experiment. Both the derivations and theoretical guarantees are presented in the paper (as well as the computational complexity in the appendix), and the effectiveness of the approach is demonstrated on a number of regression and classification tasks.
- The proposed method is architecture-independent and the authors demonstrate its effectiveness in conjunction with other meta-learning approaches such as the neural process family. In addition, seminal works on few-shot learning are shown as special cases of the hierarchical Bayesian model, demonstrating the unifying nature of this framework.

**Weaknesses:**

- The possible limitations of the approach / venues for future work are currently not discussed in the paper. Providing insights into the limitations and investigations left for future work could provide a better context for the paper in the few-shot learning community.
- Though this is potentially a matter of personal preference, the flow of the paper could be improved by putting the toy experiment either before the derivations or after the theoretical analysis. Otherwise, it seems a bit out of place. Moreover, adding a proof sketch for Theorems 5.1 and 5.2 in the main paper would be helpful even though the full proofs are provided in the appendix.

**Questions:**

- Could distributions other than NIW be considered for the prior and variational posterior? What are the limitations of using NIW? In other words, what limitations does the choice of the prior impose on the effectiveness of the approach and how does it compare to existing few-shot learning frameworks?

---

> ### Author Response · Authors · 2023-11-20
> **Responses to Reviewer wcX1**
>
> >  **1. Possible limitations of the approach for future work.**
>
> * **Limitations**: 1) NIW-Meta introduces some extra hyperparameters (eg, the number of SGLD iterations, the number of burn-in steps). These are currently estimated empirically, but a more rigorous study on how to select them automatically needs to be addressed. 2) Although it is empirically verified that our quadratic episodic loss optimisation is effective, more theoretical analysis on the quality of this approximation as well as its impact on the final results, needs to be done.
>
> * **Future works**: We have quite an extensive evaluation on popular few-shot classification and regression benchmarks. However, we would like to evaluate our approach on new emerging applications of FSL such as efficient learning of implicit neural representations such as NERF, eg [A].
>
> *[A] Tancik et al, CVPR21, Learned initializations for optimizing coordinate-based neural representations.*
>
>
> > **2. Layout of the toy experiment section. Adding a proof sketch for Theorem.**
>
> * **Toy experiment section**: We can move the section after the theoretical analysis section. We will think about which way is the best for presentation.
>
> * **Proof sketch for theorems**: We only put the final theoretical results in the main paper due to the lack of space. But we agree with the reviewer that it would be helpful to provide some overview or sketch of the proofs. We will figure out how we can include them concisely should the paper be accepted.
>
>
> > **3. Choice of the NIW distribution. Limitations of using NIW.**
>
> The reason for employing NIW is mainly because it enables closed-form solutions to local episodic optimisation, which is essential for fast task adaptation in episodic FSL.
>
> A Potential limitation of NIW might be that NIW is a single-modal distribution, thus of limited flexibility compared to the unrestricted, say nonparametric distribution families. For example, if mixing classification (object recognition) and regression (pose estimation) episodes which might prefer position invariant and position equivariant feature extraction respectively, then a uni-modal prior might be overly restrictive and multi-modal prior might be preferred. Still, it’s unclear how to perform *efficient* task adaptation with a multi-modal prior.
>
> **Impact of the NIW assumption compared to other FSL methods**: As we mentioned in Sec.3.1 (Interpretation), those MAML, ProtoNet and Reptile are all shown to be special cases of our framework, even when we followed our NIW prior-posterior family. So there is no substantive limitation compared to existing FSL methods because of our NIW family choice.

---

> > ### Comment · Reviewer_wcX1 · 2023-11-22
> > **Thank you!**
> >
> > I thank the authors for the clarifications! I suggest that the limitations and future work mentioned by the authors are included in the revised manuscript.

---

> > > ### Author Response · Authors · 2023-11-23
> > > **Thank you very much!**
> > >
> > > Thank you very much for the suggestions! We have added the limitations and future works in the revised manuscript.

---

### Author Response · Authors · 2023-11-20
**Revised manuscript with additional experiments and clarifications**

We thank all reviewers for their insightful and constructive comments/questions. Below we highlight some of the commonly asked comments by the reviewers and our responses:


**1. The confidence intervals of empirical results that are missing in some tasks (esp., Table 3).**

Now we have added the SEMs for the results in Table 3 to show the statistical significance. Please see the revised paper for the full results and also the table below for the DINO/s CIFAR case. Even though for some tasks there are some overlaps of the error intervals between our proposed approach and the competing methods, overall our approach is superior to the existing methods with statistical significance to some extent.

| CIFAR, DINO/s | 1-shot | 5-shot |
| ------------- |:-------------:| -----:|
| ProtoNet      | $81.1 \pm 0.29$ | $92.5 \pm 0.13$ |
| MetaOpt      | $70.2 \pm 0.22$ | $84.1 \pm 0.27$ |
| MetaQDA    | $77.2 \pm 0.34$ | $90.1 \pm 0.18$ |
| **NIW-Meta (Ours)** |  $\bf{82.8 \pm 0.26}$ | $\bf{92.9 \pm 0.11}$ |


**2. Uncertainty calibration by the test marginal likelihood measure, and comparison to Bayesian Neural process models.**

As requested by some reviewers, we have now done new experiments to evaluate uncertainty calibration in terms of the predictive marginal likelihood at test time. On the SineLine dataset we report the predictive log-marginal-likelihood scores (LMLHD). We try to match the experimental settings from (Volpp et al., ICLR’23) so that the results are comparable. More specifically, we test on 256 tasks, each of which consists of 64 samples with a varying number of context samples (1,2,4,6,8,12, and 16). The number of posterior samples used to compute/approximate the predictive marginal likelihood is 1024. The results are as follows (the higher the better uncertainty quantification and generalisation). As shown, our NIW-Meta has higher scores, especially for larger context size, implying that capturing uncertainty in full model parameters is important for generalisation capability.

| Context size | 1 | 2 | 4 | 6 | 8 | 12 | 16 |
| ------------- |-----|-----|-----|-----|-----|-----|-----|
| (Volpp et al., ICLR’23) | -18.38 | -15.98 | -13.69 | -12.74 | -11.75 | -11.14 | -10.23 |
| NIW-Meta (Ours) | -17.02 | -14.74 | -12.60 | -8.14 | -3.51 | -1.55 | -1.33 |

---

### Meta-Review · Area_Chair_Pzdw · 2023-12-08

**Metareview:**

This paper proposes a new hierarchical Bayesian model for the few-shot meta learning problem. It models the uncertainty of all model parameters and apply a Normal-Inverse-Wishart model on the global and local random variables with a quadratic approximation for efficient inference.

**Strengths**:
- Full Bayesian approach to all global and local model parameters
- Efficient computation compared to other methods that require repeated full model computation in the inner loop
- Rigorous PAC-Bayes theoretical analysis
- Good empirical performance on both accuracy and uncertainty calibration

**Weaknesses**:

Most concerns from reviewers have been addressed in the rebuttal. It would be important for the reviewers to incorporate their responses to the final revision. What remains is the question on the empirical evaluation on the quadratic approximation from reviewer PGLR. It would be useful for the authors to discuss its potential weakness.

**Justification For Why Not Higher Score:**

The topic of hierarchical Bayesian approach to few-shot meta-learning is extensively studied. The proposed method has its merit in an efficient solution with a SOTA performance in accuracy and uncertainty estimation, but has relatively limited novelty compared to existing works.

**Justification For Why Not Lower Score:**

Good motivation, well execution of the idea, theoretical support, and good experiment results.

---

### Decision · Program_Chairs · 2024-01-16

Accept (spotlight)